

# Towards Interpretable LSTM-based Modelling of Hydrological Systems

Luis A. De la Fuente[1], Mohammad Reza Ehsani[1], Hoshin V. Gupta[1], and Laura E. Condon[1]

Department of Hydrology and Atmospheric Sciences, The University of Arizona, Tucson, 85721, United State

*Correspondence to*: Luis A. De la Fuente (ldelafue@arizona.com)

**Abstract.** Several studies have demonstrated the ability of Long Short-Term Memory (LSTM) machine learning based modeling to outperform traditional spatially lumped process-based modeling approaches for streamflow prediction. However, due mainly to the structural complexity of the LSTM network (which includes gating operations and sequential processing of the data), difficulties can arise when interpreting the internal processes and weights in the model.

Here, we propose and test a modification of LSTM architecture that represents internal system processes in a manner that is

analogous to a hydrological reservoir. Our architecture, called HydroLSTM, simulates behaviors inherent in a dynamic system, such as sequential updating of the Markovian storage. Specifically, we modify how data is fed to the new representation to facilitate simultaneous access to past lagged inputs, thereby explicitly acknowledging the importance of trends and patterns in the data.

We compare the performance of the HydroLSTM and LSTM architectures using data from 10 hydro-climatically varied catchments. We further examine how the new architecture exploits the information in lagged inputs, for 588 catchments across the

USA. The HydroLSTM-based models require fewer cell states to obtain similar performance to their LSTM-based counterparts. Further, the weights patterns associated with lagged input variables are interpretable and consistent with regional hydroclimatic characteristics (snowmelt-dominated, recent rainfall-dominated, and historical rainfall-dominated). These findings illustrate how the hydrological interpretability of LSTM-based models can be enhanced by appropriate architectural modifications that are physically and conceptually consistent with our understanding of the system.

**1 Introduction**

Scientific research that incorporates Machine Learning (ML) has exploded over the last several years, and Hydrology is not an exception. Reasons for this include the existence of open-source APIs, the availability of large dataset repositories, and the ability to obtain good performance without requiring too much computational power (Hey et al., 2020; Pugliese et al., 2021). However, understanding in a hydrologic context what is happening inside such models continues to limit the interpretability of their results

(Nearing et al., 2021).

Reasons for this lack of interpretability are diverse, but one of the most fundamental reasons is that many of the ML architectures have been developed to address problems that are, in many respects, quite different from the ones relevant to Hydrology and/or the Earth Sciences. Specifically, many were developed in the general field of Data Science with a specific focus on classification or predictive performance, rather than on knowledge extraction. In contrast, the scientific method typically presumes some degree

of "interpretability/understanding" in the formulation of hypotheses and experiments. Using ML-based approaches for hypothesis testing can be challenging if we are unable to interpret what is happening inside the model or what is learned by our representations, and how they are related to our scientific questions. We consider interpretability to be "*the degree to which an observer can understand the cause of a decision*" (Miller, 2019), which is fundamental if we want to learn from the analysis of data.



### 1.1 Lack of LSTM Interpretability

In Hydrology, the Long Short-Term Memory (LSTM, Hochreiter & Schmidhuber, 1997) architecture has exhibited excellent predictive performance in multiple areas such as the prediction of streamflow (Kratzert et al., 2019), water temperature (Qiu et al., 2021), water table levels (Ma et al., 2021), and snowpack (Wang et al., 2022). However, while it has become one of the default algorithms used in any new ML-based hydrology research, much of this development has not been accompanied by discoveries that expand on the existing hydrology knowledge base.

The use of many sources of data (dynamic and static) as input, hundreds of cell-states (neurons or state variables), and large numbers of trainable weights (as many as thousands or even millions) in the construction of the internal representation help to ensure that the task of extracting interpretable knowledge from a trained model becomes almost impossible. A review of the hydrological literature shows that many LSTM-based streamflow prediction studies have used between 20 to 365 cell-states or more (Kratzert et al., 2018; Gauch et al., 2021) depending on the catchments trained and the depth of the network, which makes

the problem of interpreting the information contained within those cell-states a complex task. In contrast, most spatially lumped water-balance models (conceptual models) have the order of only two to six state variables (e.g., see GR4J, Perrin et al., 2003, and SAC-SMA, Burnash et al., 1973, respectively). It is possible, therefore, that either the corresponding LSTM-based models are not efficient (parsimonious) representations of the input-state-output dynamics, or that our conceptual hydrological models are overly simplified representations of reality (over-compression). In this paper, we make an argument for the former explanation.

### 1.2 Previous Work on the Interpretability Issue

Considerable effort has been devoted to understanding the nature of the relationships learned by ML-based models and summarizing the techniques available for doing so (Molnar, 2022; Carvalho et al., 2019; Linardatos et al., 2020). Some of these techniques are generic (model-agnostic), such as the use of permutation feature importance and partial dependence plots (Friedman, 2001). Others are model-specific to particular ML methods such as those that exploit the information provided by the

backpropagation of gradients (gradient-based methods). The permutation feature importance approach (Breiman, 2001) is based on quantifying the improvement/deterioration in performance when a given feature is included/excluded from the data used as input. This can be very useful for understanding the overall sensitivity of the output to the properties of the input and output, but the same analysis cannot be easily performed for specific events. On the other hand, the Expected Gradient approach (Erion et al., 2021) can be used to score the importance of a specific realization of the input using an integrated gradient over a predefined path.

However, the task of generalizing from this information requires the analysis of a large number of representative cases. These characteristics limit the ability to interpret the underlying system, and a method that is more generally able to extract knowledge from the data would be desirable.

Nonetheless, some remarkable uses of the above-mentioned methods for Hydrological investigation have been reported. Addor et al. (2018) ranked the importance of traditional static attributes in 15 traditional hydrological signatures, to obtain useful insights

into the role that static attributes play in determining the nature of the input-output relationship. Jiang et al.  (2022) analyzed the gradients in an LSTM-based model during flooding events and defined three characteristic input-output mechanisms (snowmelt, recent rainfall, and historical rainfall dominated) that facilitate an understanding of the roles that relevant attributes and dynamic forcings play in streamflow prediction, and how they can be interpreted in the context of existing hydrological knowledge.

Other efforts to interpret the results of LSTM-based representations have included the incorporation of physical constraints such

as mass conservation (Hoedt et al., 2021), feature contexts in some of the gates (Kratzert et al., 2019), post-analysis of the states (Lees et al., 2022), and use of ML-based models coupled with conceptual models (Khandelwal et al., 2020; Cho and Kim, 2022; Cui et al., 2021). However, these previous approaches have not explicitly exploited the isomorphism between the structures of the



LSTM and that of conceptual hydrological models to show how the learned weights (parameters) can be informative regarding the nature of the underlying hydrology processes.

**1.3 Objectives and Scope of this Paper**

Our goal is to enhance the interpretability of ML-based models of dynamical hydrological systems. Section 2 discusses the similarity between equations of the LSTM and the hydrologic reservoir, Section 3 uses these insights to propose a new LSTM-like architecture (called HydroLSTM) that can be hydrologically interpreted, and Section 4 discusses our general experimental methodology. In Section 5 we discuss an experiment that compares the performance of HydroLSTM-based and standard-LSTM-

based models trained to simulate the input-state-output behaviors of 10 carefully selected catchments located in differing hydroclimatic regions. Based on those results, Section 6 examines how the new architecture performs over a larger dataset of 588 catchments and discusses the implications of the creation of a single "global" model. Finally, Sections 7 and 8 discuss the benefits obtained by using parsimonious, carefully designed representations, in terms of the potential for enhanced hydrological interpretability.

**2 Structural analysis of LSTM**

**2.1 Structure of the LSTM**

Hochreiter & Schmidhuber (1997) proposed the LSTM representation as a solution to the "exploding and vanishing gradients" problem that can occur during backpropagation-based learning using recurrent networks. This problem can occur when there exist long-lagged relationships between inputs, in other words, when the system state at the current time step depends on conditions

from some distant past, as can occur in hydrological systems.

Note, however, that the meaning of what is understood as "short" and "long" memory can differ from field to field. In Hydrology, we commonly understand catchment "memory" as referring to some kind of within-catchment storage of information that influences how its behavior in the current time step depends on events (such as meteorological forcings) occurring in the past (de Lavenne et al., 2022). In this paper, we will refer to short-term memory as that where the influence of the past system inputs only

extends to a few weeks (or perhaps a season), and long-term memory as that where the influence can extend to the indefinite past (potentially many years), typically through the storage of water in the catchment (in the forms of groundwater, lakes, soil moisture, and snowpack, etc.). To be clear, these specific hydrological conceptions of memory may, or may not, align with those associated with the use of the standard LSTM representation or in other fields (such as natural language processing).

Regardless, as has been amply demonstrated, the LSTM architecture is well suited to generating predictions of the behaviors of

complex dynamical hydrological systems (Kratzert et al., 2019; Qiu et al., 2021; Ma et al., 2021; 2022). This is mainly due to its abilities to (i) represent Markovian behavior through its "*cell states*", and (ii) its ability to learn the functional forms of the "*gating*" mechanisms that determine what kinds of information are retained (or forgotten) at each time step (Lees et al., 2022; Kratzert et al., 2019). Specifically, the forget gate (denoted by the symbol $f$) can control how conservative the system is during the year (e.g., rates of water loss from storage can be greater during the summer than in the winter). Similarly, the input gate (denoted by the

symbol $i$) can control how much information is added to the system (e.g., for the same rates of daily precipitation and potential evapotranspiration, the amount of available water can be different in summer versus winter due the plant varying uptake). Finally, the output gate (denoted by the symbol $o$) can control the fraction of the system state that is converted into output at any given time (e.g. irrigation demand can change the diversion of water from the river so that different values of the streamflow can be observed for the same condition of soil moisture in the catchment). In other words, the gating mechanism enables (at each time





step) the dynamical storage and updating of information that is relevant to generating the prediction of interest. This ability of LSTM-based models to track and exploit both past and current information enables it to successfully emulate the behaviors of complex dynamic systems (Kratzert et al., 2019).

**2.2 Similarities with the Hydrology Reservoir**

To understand what is happening "*under the hood*" of the standard LSTM formulation, it is instructive to compare its structure and
function to that of the so-called "*hydrological reservoir*" model in hydrology. This can be thought of as the simplest structural component underlying the development of many conceptually understandable input-state-output models of dynamical physical systems (mass and/or energy conserving).

Consider, for example, the (so-called) "*linear reservoir*" model (Table 1), in which the precipitation excess enters a bucket where it is stored until it is released. The amount of release is related to the volume of water that is present in the bucket at each time step.
In the case of a linear relationship between storage and streamflow h=o*S(t). When o is a time-constant value, and the system equations can be solved analytically. However, this relationship can be nonlinear and/or depend in a more complex manner on the system state or the time history of inputs (i.e., o=O(S(t)) can vary with time). In this more general case, the system equations are commonly solved via numerical integration, the simplest being the explicit Euler approach, which results in the difference equation commonly used to track the time evolution of the water storage S(t) (see Table 1).
In many ways, the structure underlying a cell-state of the LSTM architecture is isomorphic to that of the hydrological reservoir after the application to the latter of finite difference approximation of the ordinary differential equation. Table 1 (adapted from L. De la Fuente, 2021) shows that the input, output, and forget gates in the LSTM represent scalar dynamical functions (asymptotic to 0 and 1). These correspond to the scalar constant-valued conductivity coefficients used in the linear reservoir, in the sense that each controls the "*flow rate*" of time-variable quantities into and out of the corresponding cell state. So, while the LSTM functions
g and $\bar{\bar{c}}$ represent non-linear transformations, they can be understood to be identity functions in the case of the linear reservoir. In other words, the LSTM gate operators and transformations cause it to behave analogously to a non-linear reservoir.

Going further, the variable S in the linear reservoir formulation represents the aggregate "*physical state*" of the system, which can be comprised of multiple physically interpretable components such as snow accumulation, moisture in different parts of the soil system, storage in the channel network, etc. Similarly, variable c in the LSTM formulation represents the aggregate "*informational*
*state*" of the system, which can be comprised of multiple components that are relevant to the predictive task at hand.

**Table 1: Comparison between Linear Reservoir and LSTM**

| Linear Reservoir | LSTM |
|:---:|:---:|
| $\dfrac{dS}{dt} = x - h$ | $\dfrac{dc}{dt} = g(x, h)$ |
| $S: water\ storage$ | $c: information\ storage$ |
| $x: input$ | $x: input$ |
| $h: output$ | $h: output$ |
| $S(t) = f \cdot S(t-1) + i \cdot (x - h)$ | $c(t) = f \cdot c(t-1) + i \cdot g(x, h)$ |
| $f = 1$ | $f = f(x, h)\ \ ]0,1[$ |
| $i = 1$ | $i = i(x, h)\ \ ]0,1[$ |
| $h = o \cdot S(t)$ | $h = o \cdot \bar{\bar{c}}(t)$ |
| $o = ]0,1[$ | $o = o(x, h)\ \ ]0,1[$ |
| | $\bar{\bar{c}}(t) = \bar{\bar{c}}(c(t))\ \ ]-1,1[$ |



Importantly, this informational state of the LSTM can be regularized to obey conservation (mass, energy, or any other entity) by ensuring that its inputs are properly normalized and handled (Hoedt et al., 2021). However, in the more general sense, any source

of informative data (such as precipitation, temperature, radiation, wind speed, humidity, static attributes, etc.) can be used to drive the evolution of the cell states. This multisource nature of the data ingestible by an LSTM model both improves its predictive power and complicates the interpretability of what the cell states are storing.

### 2.3 Differences with the Hydrologic Reservoir

Despite the aforementioned structural similarities, there are also some differences between an LSTM cell and the hydrological

reservoir, such as how the state variable is tracked, and how context dependence informs the behaviors of the gates.

#### 2.3.1 Tracking the Evolution of the State

Some of the first applications of LSTMs were in the context of speech recognition (Graves et al., 2004) and natural language modeling (Gers and Schmidhuber, 2001). In these areas, two primary assumptions are typically applied that may not hold in the dynamic environmental system: a) a finite relevant sequence length (finite memory time-scale), and the consequent possibility of

b) a non-informative system state initialization. These assumptions can create challenges when applying LSTM to hydrologic systems.

In linguistics, the most recent symbols (letters and/or words) provide valuable context that is useful for the upcoming ones. The idea is that symbols that have previously appeared many sentences or paragraphs earlier will typically provide less contextual information than more recent ones. This standard LSTM formulation, therefore, assumes that some finite number of sequentially

ordered previous symbols will contain all (or most) of the relevant information required to establish the current context, which is then summarized by the cell- states of the model. Accordingly, all the symbols that are further away in the past than some specified sequence length are ignored when computing the cell-states, which allows for them to be initialized to zero (or some non-informative random value) at the beginning of the sequence. In other words, any linguistic activity that occurred before the assumed sequence length is treated as irrelevant to the predictive task at hand.

While this structure may make sense when dealing with linguistic applications, the assumption of current contextual information being dependent on a finite length of history does not, in general, align with how predictive context is established in a dynamical environmental system. For example, the information stored in mass-related hydrological state variables (e.g., the water content in the soil, groundwater levels, snowpack, etc.) can often be the consequence of a very long history of conditions and events that have occurred in the past. So, whereas in certain situations, it might make sense for a relevant state variable to depend on only a finite-

length history of past events (e.g., ephemeral snowpacks may be only informative about conditions since the onset of sufficiently cold weather for precipitation to occur in the form of snow), in many other situations the current wetness/energy state of the system may depend on an effectively indeterminate sequence length.

In such a system, where long-term memory effects can be important, the use of the standard LSTM assumption of a finite relevant sequence length could mean that valuable historical information present in the data is not optimally exploited. One way to account

for that information would be to implement an informative (e.g. non-zero) initial condition when implementing the LSTM with a finite sequence length. Another way would be to extend the sequence length so that errors in the initialization of the cell states are rendered minimal/unimportant. Clearly, for dynamical environmental systems, the issues of cell-state initialization and selection of relevant sequence lengths are coupled. Without addressing this potential source of information loss, a model based on the traditional LSTM architecture would be incapable of exploiting information regarding longer-term dependencies on conditions



that predate the specified sequence length; for example, the long-term decadal and multidecadal dependencies that can affect the evolution of the North American Precipitation system.

Having said this, the use of a fixed sequence length can be extremely useful for model development, as it facilitates the randomization of data presented to the model during the training process, which helps to ensure better training results and superior generalization performance. Proper randomization ensures that the data used for training and testing are statistically representative

of the full range of environmental conditions that the model is expected to provide reliable predictions (Chen et al., 2022; Guo et al., 2020; Zheng et al., 2018). In contrast, when developing models of the dynamical evolution of environmental systems with potentially long memory time scales, a more suitable formulation is one in which the effective memory length can be both variable and indeterminately long as needed (Zheng et al., 2022).

### 2.3.2 Gating Behavior

Another difference between the LSTM and hydrological reservoir structures relates to the behaviors of the gates. Whereas the input gate of the hydrological reservoir is typically an identify function, and the output and forget gates (o and f) are typically assumed to depend mainly on the current state of the system, the three corresponding LSTM gates can vary dynamically in a manner that is controlled by the system inputs and outputs.

Considering the examples mentioned in Section 2.2, where we discussed the possibility of seasonal hydrological patterns/trends

affecting the hydrological behavior of the gating mechanisms; it seems highly improbable that a gating representation based on knowledge of only the current system state, inputs, and outputs would be able to learn how to exploit relevant information about such patterns. While it is possible to implement the standard LSTM architecture in such a manner that it can use sequences of past-lagged input and output data (up to some pre-determined sequence length) to also influence the operations of the gates, this can further complicate the problem of interpretability, by making it more difficult to disentangle the relationships between the sequence

length, learned gating context, and the number of cell-states needed.

### 3 Proposed HydroLSTM structure

To address the aforementioned issues, we propose an alternative LSTM-like architecture (hereafter referred to as HydroLSTM) that more closely aligns with hydrological understanding, while retaining the behavioral strengths of the traditional LSTM. The alternative structure continues to use the standard LSTM equations for the gates (i, o, and f), cell states (c), and outputs (h), but

makes two important changes. First, the cell states are continually updated from the beginning to the end of the available dataset while maintaining the sequential ordering of the input drivers. This ensures that the cell states represent Markovian memories that are effectively of indeterminate length (as in traditional hydrological modeling, initialization is done only once at the beginning of the simulation period). Second, the gates are allowed to learn behaviors that depend on a fixed, user-specifiable, sequence of past-lagged data values that can represent (seasonal) memory of what has happened in the recent past. Accordingly, each cell state of

the HydroLSTM uses the following equations:

$$i(t) = \sigma\big(\textstyle\sum_j^{inputs} \sum_{\tau=0}^{memory} W_{i,\tau,j} \cdot x_j(t-\tau) + U_i \cdot h(t-1) + b_i\big), \tag{1}$$

$$f(t) = \sigma\big(\textstyle\sum_j^{inputs} \sum_{\tau=0}^{memory} W_{f,\tau,j} \cdot x_j(t-\tau) + U_f \cdot h(t-1) + b_f\big), \tag{2}$$

$$o(t) = \sigma\big(\textstyle\sum_j^{inputs} \sum_{\tau=0}^{memory} W_{o,\tau,j} \cdot x_j(t-\tau) + U_o \cdot h(t-1) + b_o\big), \tag{3}$$

$$g(t) = \tanh\big(\textstyle\sum_j^{inputs} \sum_{\tau=0}^{memory} W_{g,\tau,j} \cdot x_j(t-\tau) + U_g \cdot h(t-1) + b_g\big), \tag{4}$$

$$c(t) = f(t) \cdot c(t-1) + i(t) \cdot g(t), \tag{5}$$



$$h(t) = o(t) \cdot tan\,h\big(c(t)\big), \tag{6}$$

where σ represents the sigmoid function, tanh is the hyperbolic tangent function, Wand U are trainable weights, and b is a trainable bias term. All of the elements in the equations are vectors, where W, b, and x have the dimension of the number of dynamic inputs, and the functions i(t), f(t), o(t), c(t), and h(t) are scalars for a single cell state or vectors in the case that more than one cell in parallel is specified by the user). The symbol τ represents the lagged previous time steps, and j indicates the number of inputs (e.g. meteorological forcings). The "memory" term is a hyperparameter that specifies the sequence length used for determining gating behavior.

Figure 1 provides a conceptual illustration of the architectural differences between the LSTM and the HydroLSTM. The specific LSTM representation shown here (Fig. 1a) implements a sequence-to-one input-output mapping, to better match the representation used in traditional hydrologic modeling (whereas the LSTM structure also facilitates a sequence-to-sequence mapping, it does not facilitate a ready comparison). The figure highlights how the LSTM representation must be evolved from an initialized cell state (typically zero) over some specified sequence length of past data ($\bar{C}_3 = 0$) each time a prediction is required (e.g. $\bar{C}_4 = 0$ for $Q_{25}$, $\bar{C}_5 = 0$ for $Q_{26}$, and so on). This formulation limits the effective memory of the system to be shorter than or equal to the specified sequence length. The learned weights that determine gating behavior at each time step (arrows in the figure) remain constant across the sequence length (e.g. represented by the same size arrow for each time inside of the sequence length). Note that the gating dynamics are only controlled by data from the current time step. Further, the same input data values must be processed several times when learning the correct values for successive current cell states.

In contrast, while using the same input data, the HydroLSTM representation (Fig. 1b) initializes the cell-states only once at the beginning of the time-ordered data set (typically $\bar{C}_0 = 0$, but could be set to some other user-determined value). Accordingly, the state value $C_{24}$ is updated based on $C_{23}$, and so on. In this representation, the specified sequence length is used only within the gates to determine their gating behavior. We will show later (Section 5.1.3) that analysis of the corresponding learned weights (wider arrows) facilitates valuable hydrological interpretation.





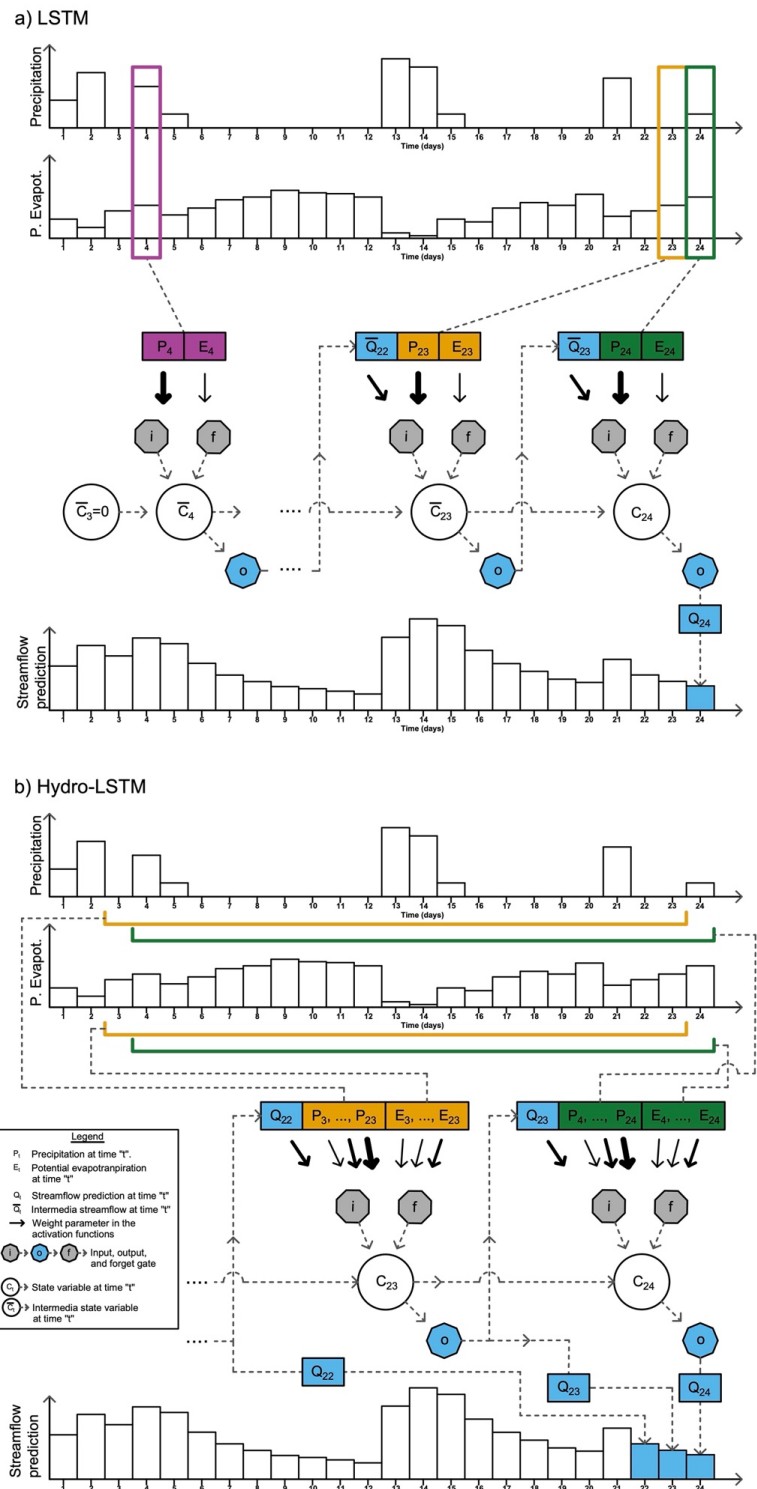

**Figure 1: Conceptual illustration of both representations. In this example, the LSTM (a) uses 20 days of past sequence data to determine the cell state value, while the HydroLSTM (b) uses 20 days of past sequence data to determine the gating behavior.**



## 4 Methods

We conduct two kinds of experiments to examine the behaviors of the standard LSTM and HydroLSTM representations: (1) In the first experiment, we examine three main aspects of the HydroLSTM and LSTM representation: a) architectural efficiency (parsimony) as measured by the number of cell-states, b) effectiveness as measured by predictive performance, and c)
interpretability of the learned weights of the HydroLSTM gates. (2) In the second experiment, we examine what can be learned about catchment memory time scale and its possible relationship to catchment attributes, by applying the HydroLSTM to a much larger number of catchments. Details of the experiments are provided in Sections 5 and 6. Here we present the overarching methods.

### 4.1 Data

Both sets of experiments use daily streamflow as the target output, and precipitation and temperature as meteorological data input,
taken from the CAMELS dataset (Newman et al., 2014). Daily minimum and maximum temperature information available through the extension of the original meteorological forcing developed by Addor et al. (2017) are used to compute a reference crop evapotranspiration using the Hargreaves & Samani (1985) equation as an approximation to potential evapotranspiration. While the reference crop evapotranspiration must be adjusted for land use to obtain corrected estimates of potential evapotranspiration, we did not directly apply this correction and instead allowed the adjustment to be implicitly learned by the model for each catchment.
Accordingly, precipitation and reference crop evapotranspiration are used as the drivers of streamflow generation.

The temporally extended data set is for the period 01/01/1980 to 31/12/2008, (approximately 28 water years). We split the data into Calibration, Selection, and Evaluation periods as indicated below (these are commonly referred to as Training, Validation, and Evaluation periods respectively in the ML literature).

- **Calibration** (20 water years): 01/01/1980 to 30/09/2000 (70.8% of the dataset). The first nine months (01/01/1980 to
30/09/1980) were used only for system initialization, and the rest was used for model (parameter) training.
- **Selection** (4 water years): 01/10/2000 to 30/09/2004 (14.2% of the dataset). This period was used to select the best-learned weights across all the epochs used in the training.
- **Evaluation** (4.25 water years): 01/10/2004 to 31/12/2008 (15% of the dataset). This period was used to assess generalization performance.

### 4.2 Machine Learning Setup

To ensure a fair analysis of the architectures, we set up the implementation in such a manner that the only differences are the architectures themselves. The ranges over which the parameters and characteristics that define this setup were adopted from traditional values commonly used in LSTM-based modeling research. Our purpose was to establish a common framework for evaluation, rather than to design some "best possible setup" which could be tweaked to favor one or other of the representations.
As such, we implemented the following.

- Only one hidden layer was used, with the number of cell-states in that layer being the hyperparameter to be explored. All of these cell-states were fully connected with the input and the output layers. The number of data lags for each input variable used to construct the input layer was also treated as a hyperparameter to be explored.
- Uniform Glorot initialization of all weights (and bias terms) was used (Glorot and Bengio, 2010), as is suitable for
networks when sigmoid functions are involved.



- Both architectures were implemented using the same (sigmoid) activation functions. We did experiment with other activation functions, but do not report those results here. More research is required to determine whether specific activation functions are particularly suitable for hydrological applications.

- The training was conducted for 512 epochs, from which the weights and biases were selected as those achieving the lower validation period error at any point during training. Based on preliminary catchment-by-catchment testing, the batch size was fixed to be 8 days.

- Parameter optimization was conducted using the stochastic Adam optimizer (Kingma and Ba, 2017) with a learning rate fixed at 0.0001.

- The loss function used was the SmoothL1norm (Huber, 1964), which combines an L2 norm for a value lower than a specific threshold and an L1 norm for values higher than that. The reason is that this norm is less sensitive than Mean Square Error (MSE) to outliers which helps to prevent exploding gradients.

- Because we seek a parsimonious representation, dropout was not implemented. Instead, stochasticity was achieved using an ensemble of 20 runs for each hyperparameter setting.

- A "*mean normalization*" procedure was implemented by subtracting the mean and dividing it by the range for each variable.

- Testing period performance evaluation was performed using the Kling-Gupta Efficiency metric (KGE, Gupta et al., 2009).

Streamflow prediction models, based on the LSTM and HydroLSTM architectures, were developed and trained 'locally' for each of these catchments (including parameters and hyperparameter combinations) In other words, only local catchment data was used for model development, and those models, therefore, represent the best possible predictive performance achievable at those 290 locations using those architectures, without access to potentially useful information from any of the other catchments.

Note that important elements of the training procedure are data selection and stochasticity. For the LSTM, a random selection of subsamples in each batch used during training helps to achieve a rapid convergence (Kratzert et al., 2018; Song et al., 2020), so the standard procedure was used. In contrast, for the HydroLSTM, the data must be fed sequentially, so the data in each batch are sorted in this way.

Further, the training of both representations is sensitive to the initialization of the parameters, so 20 different random parameter initializations were implemented for each hyperparameter combination. The mean performance achieved for those 20 different models was taken to be representative of the distribution of performance.

Given that the HydroLSTM structure requires sequential processing of the data, the corresponding state variable c(t) must be stored and used across all the time series even between batches and epochs. Consequently, the initialization of the cell states at beginning 300 of the training could be to some arbitrary value (zero or a randomly selected one). However, when iterating over multiple epochs (using the same training dataset), the cell states were initialized to their values obtained at the end of the time series, these being suitable initialization values for the next epoch in keeping with the fact that hydrological conditions at the beginnings and ends of water years tend to be similar.

## 5 Experiment 1

In this experiment, we compare the architectural efficiency (parsimony as measured by the number of cell-states) and effectiveness (as measured by predictive performance) of the HydroLSTM and LSTM over the ten catchments selected. For purposes of discussion, we show here the analysis for only two catchments from different hydrological regimes; in **Appendix A** the figures for all ten catchments are included.



### 5.1 Methodological details

Ten catchments with different hydrometeorological behavior were selected from the CAMELS dataset to perform a comparison of the LSTM and HydroLSTM architectures. Two catchments were selected to represent each of the homogeneous regions identified by Jiang et al. (2022) based on behaviors learned by an LSTM-based modeling approach applied to flow peak prediction (see Table 2 and Figure 2).

**Table 2: Information on the catchment selected.**

| Code | Name | Latitude | Longitude | Area (km2) | Criteria |
|---|---|---|---|---|---|
| 11523200 | Trinity River above Coffee Creak, near Trinity Center, CA | 41.11126 | -122.70558 | 382.94 | Recent rainfall-dominant (West) |
| 11473900 | Middle Fork Eel River, near Dos Rios, CA | 39.70627 | -123.32529 | 1925.01 | |
| 9223000 | Hams Fork below Pole Creek, near Frontier, WY | 42.11049 | -110.70962 | 333.15 | Snowmelt-dominant |
| 9035900 | South Fork of Williams Fork, near Leal, CO | 39.79582 | -106.03057 | 72.84 | |
| 6847900 | Prairie Dog Creek above Keith Sebelius Lake, KS | 39.76985 | -100.10078 | 1536.19 | Mixed |
| 6353000 | Cedar Creek, near Raleigh, ND | 46.09167 | -101.33374 | 4526.51 | |
| 2472000 | Leaf River, near Collins, MS | 31.70694 | -89.40694 | 1927.13 | Historical rainfall-dominant |
| 5362000 | Jump River at Sheldon, WI | 45.30803 | -90.95652 | 1477.29 | |
| 3173000 | Walker Creek at Bane, VA | 37.26818 | -80.70951 | 773.32 | Recent rainfall-dominant (East) |
| 1539000 | Fishing Creek, near Bloomsburg, PA | 41.07814 | -76.43106 | 701.78 | |


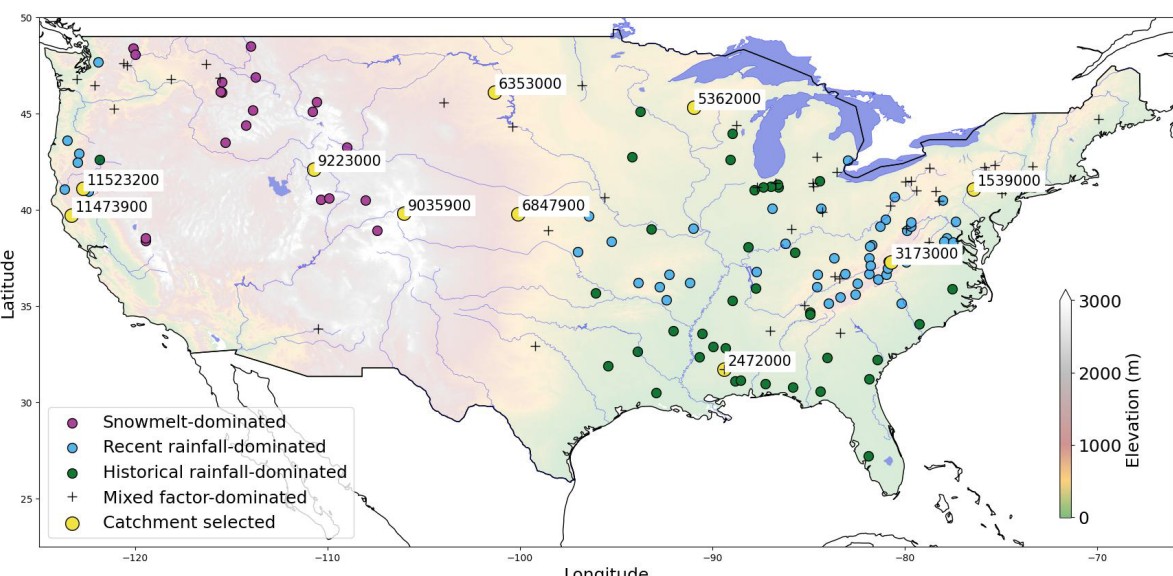

**Figure 2: Catchments selected in this study, adapted from Jiang et al. (2022). White boxes represent the catchments analyzed and the dots describe the three flooding mechanisms presented by Jiang et al. (2022).**

With the Machine Learning setup constraints defined previously, the only hyperparameters that must be tuned via grid search are
the number of cell-states and the lagged memory length (for LSTM this is the sequence length, and for HydroLSTM this is the number of lagged time steps used to determine the gating behavior). For this experiment, the numbers of cell states were varied to be 1, 2, 3, 4, 8, and 16. For the lagged-memory hyperparameter, we varied the value as powers of two (i.e., using 2, 4, 8, 16, 32, 64, 128, and 256 lagged days) to account for the fact that each extra day tends to provide decreasing amounts of information as the number of lags is increased.





In keeping with the principle of parsimony, we identified the "simplest" LSTM and HydroLSTM representations as being the ones with the smallest number of cell-states for "comparable" predictive performance. This comparison is assessed based on the mean performance over 20 realizations not being statistically distinguishable (at the $p=0.1$ significance level) with respect to one reference hyperparameter combination in HydroLSTM (red * in Fig. 3).

## 5.2 Number of cell-states

Figure 3 shows the results for two catchments, where we plot heatmaps of mean KGE performance for each combination of the number of cell states (from 1 to 16) and the number of lagged days (from 2 to 256) tested. Darker green indicates higher KGE performance (optimal KGE = 1). Each row of subplots corresponds to a catchment while the left column is for LSTM and the right column is for HydroLSTM. The cell-lag combinations for which the mean performance differences are not statistically significant are marked using the * symbol. From these, the red * shows the hyper-parameter combination used for doing the statistical analyses.

We see that for many hyper-parameter combinations, the performance is statistically similar.

For both catchments presented in Fig. 3, a HydroLSTM-based model having only a single cell-state performs (on average) as well as an LSTM-based model having a larger number of cell-states. For the Eel River, CA, (ID 11473900) both architectures obtain good levels of KGE performance (above 0.8) using 3 to 4 cell states, even though slightly better results can be achieved using 16 cell-states. This makes sense, given that the catchment is in a region where recent rainfall dominates the generation of streamflow

and where several state/storage components (such as surface, subsurface, groundwater, channel network, etc.) can be expected to be relevant to the streamflow generation process. Nonetheless, the HydroLSTM-based model with only a single cell state provides comparable performance to an LSTM-based model having two cell-states, when both are provided with the same lagged input sequence length.

Meanwhile, the South Fork of Williams, CO., (ID 9035900) is in the Front Range of the Rocky Mountains, where snow

accumulation and melt dynamics strongly govern streamflow generation. Here, the difference between the HydroLSTM and LSTM is quite marked. The HydroLSTM with a single cell-state (with 256-day lagged inputs) obtains extremely high performance (mean KGE>0.85) while adding more cell-states does not result in further statistical improvement. In contrast, the LSTM-based model requires at least 8 cell-states (with >32-day lagged inputs) to obtain comparable performance, which is a much less parsimonious characterization and makes interpretability much more difficult.

Figure 4a expands upon Fig. 3 by summarizing the results for all 10 of the catchments studied. In all cases but one, the LSTM requires more than one cell state to achieve the same performance as HydroLSTM. This is another indication that LSTM is not creating a parsimonious characterization of the input-state-output relationship. For instance, the catchments Trinity River (ID 11523200), Leaf River (ID 2472000), and Fishing Creek (ID 1539000) show the largest differences in the number of cell states between both representations, 1 cell in HydroLSTM versus 16 cells in LSTM.

**5.3 Comparison in terms of the best performance**

We next compare the best performance between HydroLSTM-based and LSTM-based models across the entire range of numbers of cell states and input lags. Fig. 4b shows the best mean KGE performance for each of the 10 catchments. The values fall close to the 1:1 line indicating similar overall performance using the HydroLSTM and LSTM architectures, which is desirable given their similarities in the structure (more details in Table B1), but with HydroLSTM consistently performing slightly better. Of course,

this is not intended to be a general conclusion, given that numerous versions of the LSTM architecture exist, and more emerge every year (e.g., sequence-to-sequence LSTMs, LSTMs with peephole connections, GRUs, bidirectional LSTMs, LSTMs with attention, etc.). It must be kept in mind that we have only tested the simplest possible LSTM structure, to be able to interpret it





using hydrological reservoir concepts. Further, the goal of this study is *not* to find a representation that outperforms the LSTM performance (or the current state of the art in recurrent neural networks), but instead to explore whether similar performance can

be obtained with a more parsimonious and physically interpretable architecture.

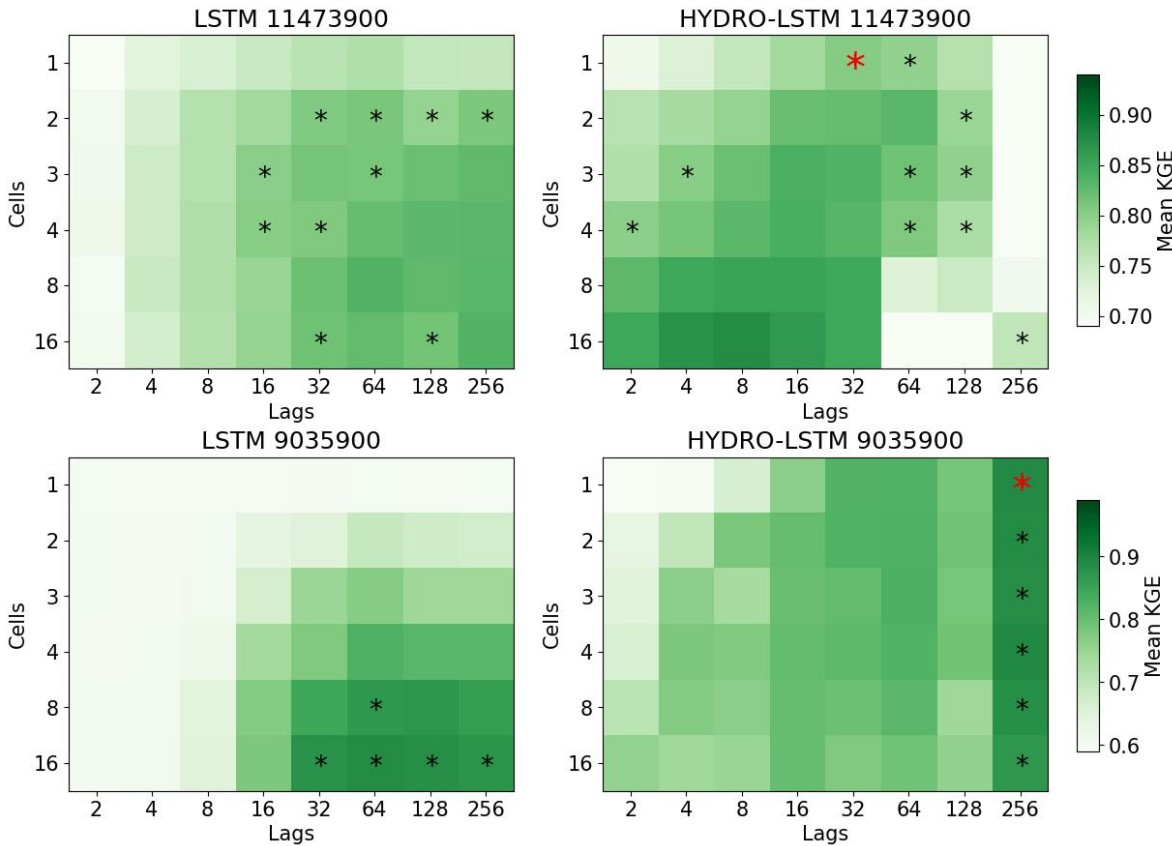

**Figure 3: Performance heatmap for different hyperparameter sets, number of cell-states and lag days. Rows: Catchment studied. Columns: representation. "*" shows the hyperparameter sets with no statistical difference in the mean with respect to the red * for each catchment. The green color (good performance) is presented closer to the row of #1cell for HydroLSTM than LSTM representation**
**which is an indication of parsimony of the former.**

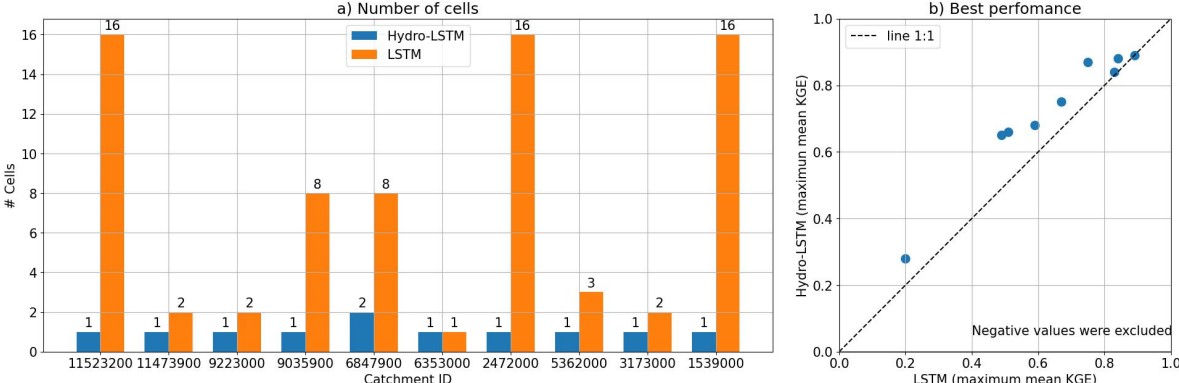

**Figure 4: Summary of cell-state parsimony and KGE performance. a) Comparison of the minimum number of cells needed for each representation having a non-statistical difference in the mean performance. b) Best performance of each representation across all the hyperparameter sets explored.**





### 5.4 The temporal pattern in the distribution of weight


For all but one of the ten catchments represented by Fig. 4a, we found that good performance can be obtained by a HydroLSTM-based model having only one cell state. This is convenient, as it allows us to compare (across catchments) the patterns of the "gate weights" learned for each model in the 20-member ensemble. For this purpose, we examine the results obtained for two selected catchments, one being rainfall-dominated and the other being snowmelt-dominated. Specifically, **Figure 5** shows the distributions

of the gate weights, associated with the lagged precipitation and potential evapotranspiration inputs, for the single-cell HydroLSTM-based models with lag-memory hyperparameter values that provide the best model performance. Despite the dispersion of weight values across the ensemble of 20 models, there are clear systematic trends in the time-lag patterns of the distribution of the weights. In general, at longer lag times (e.g., > 10 days) the weight distributions tend to encompass zero, suggesting that L-norm regularization could be used to better constrain these values during training (and that such values could be

interpreted as effectively zero). For this study, we decided _not_ to train with L-norm regularization because, rather than seeking a minimal number of weights, we are interested in the interpretability that might be associated with the time-lagged patterns seen in the trained weights, and whether these patterns might represent specific characteristics associated with different catchment "types". In this regard, we note that the output hydrological response of the Eel River CA., (ID11473900, upper row) is governed by recent-rainfall events (Table 2), which aligns well with the high weighting assigned to precipitation at time zero (~$10^{-1}$ in the figure) in

all of the gates, and particularly in the output gate that directly controls the streamflow response. The rapid decline (towards zero) in weight magnitude with time lag is consistent with a system having a relatively short hydrological memory. Further, the weights associated with potential evapotranspiration tend to be very close to zero, indicating its relative lack of importance in governing streamflow generation. These characteristics are consistent with the hydrologic classification reported by Jiang et al. (2022).

In contrast, the results shown for the snowmelt-dominated catchment (lower row of Fig. 5) are quite different. Now we see

significantly larger weight values associated with potential evapotranspiration (calculated using the Hargreens & Samani equation) which is strongly determined by air temperature, which in turn is the primary driver for snowmelt dynamics. Moreover, the weights remain at significant values for as long as 30 – 40 days, which is consistent with the time durations associated with energy/heat accumulation required for the melting process to begin resulting in a significant generation of streamflow.

Results for the other 8 catchments are presented in Appendix C. In general, the weight patterns correspond well with the

hydrological classification presented in Table 2. Models for the "western recent rainfall-dominant" catchments assign higher weights to recent precipitation, while models for "snowmelt-dominant" catchments assign high weights to about 20+ days of potential evapotranspiration (as a surrogate for temperature). Models for the "historical rainfall-dominant" catchments assign high weights to several past days of precipitation, while models for "eastern recent rainfall-dominant" catchments have weight patterns indicating longer resident times in that part of the country (eastern). In general, these results support the idea that the learned weight

patterns can encode useful information regarding the hydrometeorological characteristics of different catchments.





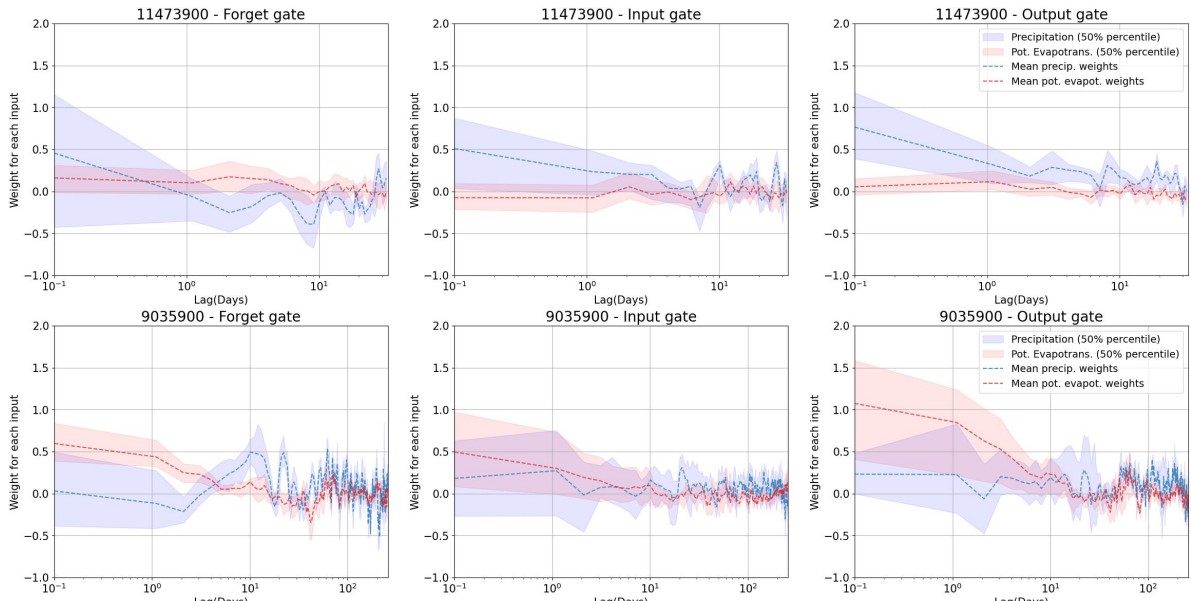

**Figure 5: Weight distribution in the three gates for two of the catchments studied. The upper row is a catchment in a recent-rainfall-dominated region. The lower row is a catchment in a snowmelt-dominated region. The confidence interval is the result of running 20 models with random initialization.**

## 5.5 Temporal patterns in the evolution of the cell state

Hoping to obtain further hydrological insights, we also examined the patterns in the temporal evolution of the (single) cell-state obtained for the trained HydroLSTM models. Specifically, we examined the results obtained for the South Fork of Williams Fork, CO, catchment (ID9035900) for which the best KGE performance (over the 10 catchments) was obtained. However, despite having only one cell state and high performance, the cell state trajectories of the ensemble of 20 models displayed no mutually-consistent trends, and therefore poor interpretability (Appendix B). Accordingly, we did not pursue such an analysis for the other catchments. We revisit this issue in the discussion section of the paper.

## 6 Experiment 2

We are interested in the performance of the HydroLSTM architecture when applied to a large sample of catchments. Here, we specifically explore how the amount of lagged memory varies geographically. Since, in the first experiment, remarkably good KGE performance was obtained (for the ten, quite hydro-climatically diverse, catchments tested) by use of a HydroLSTM architecture with only a single cell state in the hidden layer (see Fig. 4a), we proceed by fixing the number of cell-states at one (the most parsimonious choice) and use data from a larger number of catchments to explore how the optimal number of sequence time lags used for gating varies across the country. Note that an optimum number of cell-state for each catchment could be found too, however, that is a computationally expensive task that we will explore in future works.

## 6.1 Methodological details

In this experiment, we expanded the number of catchments used from the CAMELS dataset. From the 671 catchments originally available, we used the 588 catchments that have streamflow data for the entire calibration, selection, and evaluation period, and



one HydroLSTM-based model was trained for each of the catchments selected. These 588 catchments represent similar hydroclimatology diversity as the original 671.

Under the constraint of one cell-state per catchment, different settings of the 'lag memory' hyperparameter (i.e., 4, 8, 16, 32, 64, 128, and 256 lagged days) are evaluated and the performance is compared.

**6.2 Results**

The cumulative distribution (CDF) of KGE performance is presented in Fig. 6a. The CDFs shift to the right (indicating improved overall performance) as we provide the models with increased numbers of lagged inputs (consistent with increasing system memory

time scales). This seems reasonable, given that the models can access greater amounts of information regarding the history of the corresponding catchment system. However, performance improvements saturate at around 256 days of memory, consistent with the 270 days sequence length determined by Kratzert et al. (2019) to be suitable when training a single LSTM model for the entire CAMELS dataset.

However, when we independently search for the optimal sequence length associated with each catchment, we obtain the red line,

which is shifted even further to the right. This suggests that the use of a fixed sequence length (memory time scale) across the CONUS is not optimal and that better results can be obtained by allowing the sequence length to be determined along with the trainable parameters of the model. Arguably this makes sense since the system memory time scale can be expected to be a characteristic property of each catchment. Accordingly, when seeking to create a single "global" model that can be applied universally to all of the catchments used for model development, it would be desirable for the chosen representational architecture

to be sufficiently flexible to be able to learn this characteristic.

As an attempt in this direction, Fig. 7 shows the spatial distribution of the optimal sequence length determined above (corresponding to the red line in Fig. 6a). Unfortunately, while some rough regional patterns are apparent, they do not stand out clearly. For example, longer sequence lengths seem to correspond to mountain ranges (Appalachian, Cascades, and Rocky Mountains), while shorter memory seems to be associated with smaller catchments that are far from major rivers. So, while we

might expect (from a functional perspective) that sequence lengths should correspond to some distinguishable attributes of the catchments, geographic location is not sufficient for this purpose. Our results suggest that a more detailed future investigation of how the optimal sequence length (as an indicator of system memory) corresponds to observable catchment attributes could prove to be useful and informative.

Following De la Fuente et al. (2023), we also plot the optimal model KGE versus aridity index (AI) for all 588 catchments (Fig.

6b). We see that the moving average trend (over 15 catchments) and dispersion of KGE performance remain fairly stable in the energy-limited regime (AI between 0.25 and 0.6 mm/mm), from which we can infer that the input-state-output relationships in such regions are reasonably well characterized using only precipitation and potential evapotranspiration (or temperature) as the system drivers. However, for water-limited regions, as the aridity index increases, the dispersion gets larger and the average model performance declines, suggesting that these two drivers alone (along possibly with data quality issues) are insufficiently

informative to achieve good predictive performance. While it is, of course, possible that the situation could be remedied by the appropriate choice of a "better" representational architecture, the fact that De la Fuente et al. (2023) obtained similar results using three different representational approaches tends to suggest that the problem may lie with the data instead. Note also that the upper boundary of performance (~0.9) remains relatively insensitive to the aridity index, even in the water-limited regions. The fact that performance is still high in some cases having high aridity index suggests that some other catchment attribute (such as slope, area,

elevation, etc.) may also be relevant to the ability to achieve good model performance. However, it is clear that more work is required to disentangle the factors associated with overall model performance in arid regions.





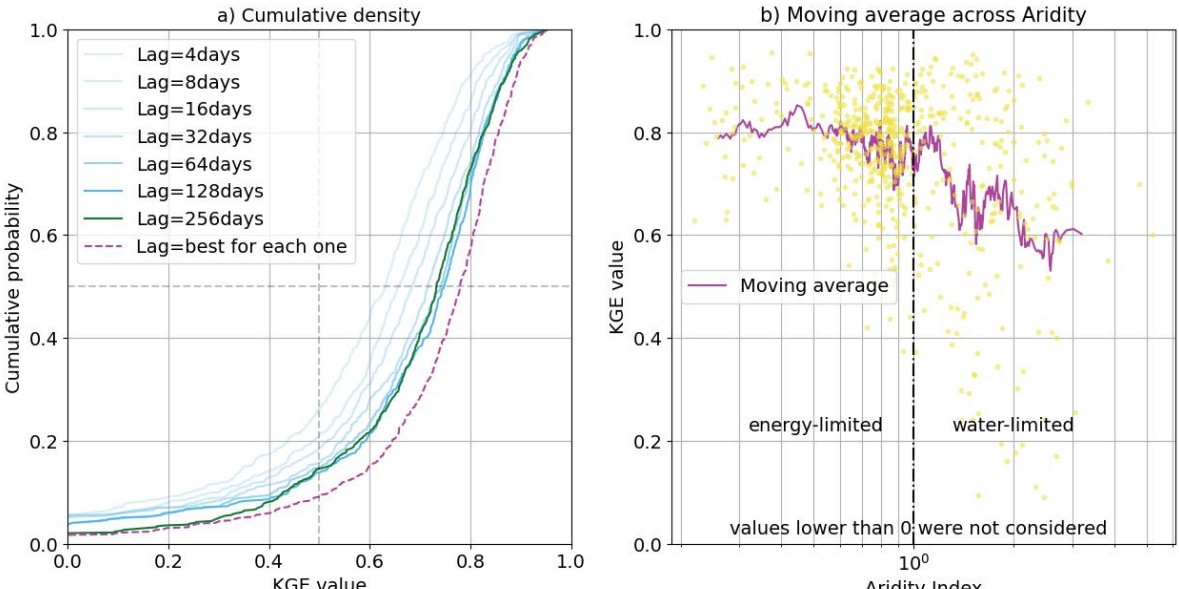

**Figure 6: HydroLSTM performance over 588 catchments (one model per catchment). a) Cumulative density function for catchments trained using different amounts of lag memory (green and blue lines) and performance for the best catchment-specific lag (red dashed line). b) Performance versus aridity index (the red line is a 15 catchment moving average) showing different behaviors for energy-limited and water-limited regions.**

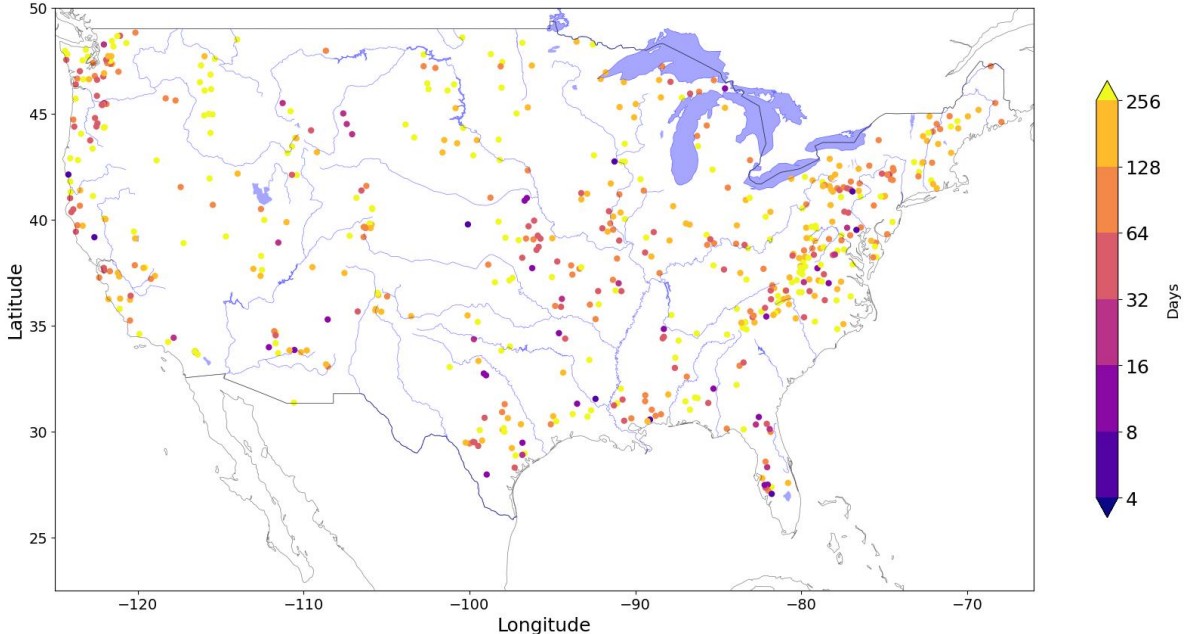

**Figure 7: Spatial distribution of the optimal lag memory determined for each catchment. No obvious pattern related to catchment attributes is apparent.**

Complementary to the findings of memory being catchment-specific (Fig. 6a) and the relationship between aridity and performance (Fig. 6b), we also examined model performance when the amount of lag memory is associated with different clusters of catchments. We hypothesize that for arid catchments, better predictive performance will typically require data with longer lags than for wet



catchments. Accordingly, we defined 4 subgroups corresponding to different levels of aridity, while maintaining a minimum level of KGE performance above 0. Further, given that long memory time scales are typically expressed via baseflow, we restrict the

analysis to catchments with a baseflow index (ratio of mean daily baseflow to mean daily discharge) above 0.5.

The results are shown in Fig. 8. We see that for catchments with AI<0.6, an optimal number of lagged days of input is around 64 (~2 months), but performance is relatively insensitive to the number of lags. For catchments with 0.6<AI<0.8 and 0.8<AI<1.0, we see more pronounced increasing trends in performance with the number of lags, with the optima being at 128 (~ 4 months) and 256 (~8 months) days respectively. Finally, the water-limited catchments (AI>1) exhibit much greater sensitivity to memory time

scales. In this preliminary study, the largest number of time lags examined was 256 days, but the results suggest that longer time scales would be worth investigating in future work. Overall, the results suggest a strong relationship between required memory time scales and aridity, which is consistent with the conclusions in De la Fuente et al. (2023), that improved representation of groundwater-related processes is required when modeling water-limited catchments.

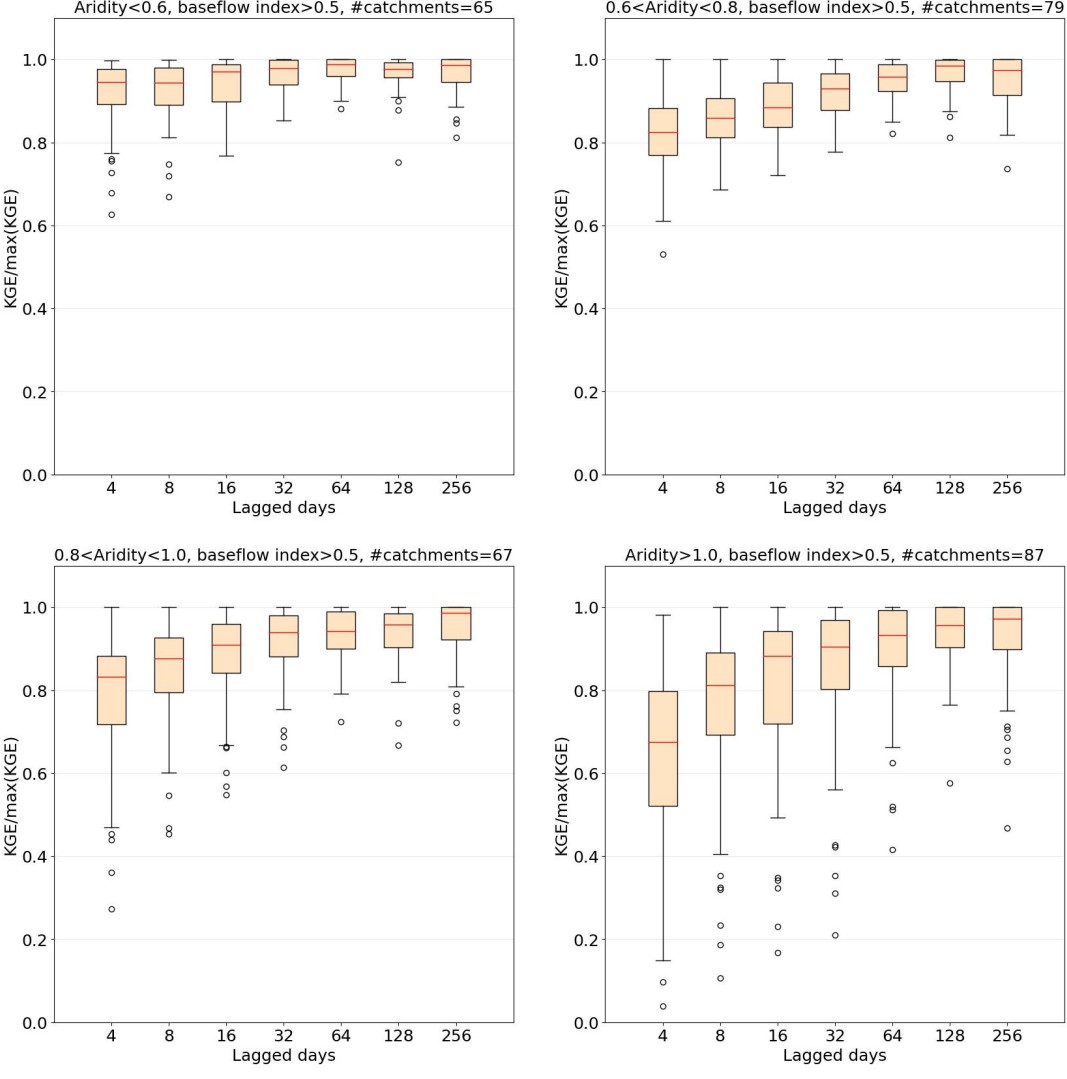

**Figure 8: HydroLSTM performance as a function of sequence length for four catchment subgroups associated with different levels of aridity. Wet (energy-limited) catchments are relatively insensitive to sequence length, while arid (water-limited) catchments require longer sequence length.**



## 7 Discussion

The main motivation for creating the HydroLSTM architecture was to explore how machine learning methodologies can better
support the development of hydrological understanding. In this section, we discuss the interpretable nature of the weights and cell-
states of HydroLSTM-based models.

### 7.1 Interpretation of the weighting pattern

[61] Due to the high dimensionality and algorithmic complexity of typical ML-based representations, the learned weights are
commonly considered to be non-interpretable (Fan et al., 2020). However, the weights that determine the behavior of each gate in
the proposed HydroLSTM architecture can, when viewed as sets, be interpreted as representing "*features*" or "*convolutional filters*"
that are applied at each time step to the sequence of lagged inputs. As such, these filters act to extract (via temporal convolution)
contextual information about the recent hydrometeorological history that can be expected to govern the current response of the
catchment. A complementary interpretation is that the filter serves as a compressed, low-dimensional, embedding of the
information encoded in the high-dimensional space of the lagged inputs and weights. In other words, the information contained in
hundreds of highly correlated lagged inputs is transformed into a small number of scalar values that succinctly express the
information needed to determine the behaviors of the gates. This "*information bottleneck*" process (Parviainen, 2010) has been
shown to perform well at dimensionality reduction and help achieve linear scaling in training time. Accordingly, the relative high
dimensionality of inputs to a gate is not a serious problem, given that the compressed (latent) space tracks only the information
required for determining catchment behaviors. In brief, the temporal patterns associated with the learned gating weights can be
informative about what is being learned by the network.

### 7.2 Interpretation of the cell state trajectories

As mentioned in the introduction, architectural parsimony (expressed as a smaller number of state variables) can lead to better
interpretability of the information encoded into a model. Here, we have demonstrated that the HydroLSTM approach enables a
representation of catchment input-output dynamics to be achieved using only a relatively small number of cell-states per catchment.
However, even when we use only a single cell-state to represent the storage dynamics of a catchment, this does not ensure that a
unique solution.

The reasons for this non-uniqueness are worth considering. It is important to note that the high model performance (as in section
5.5) only ensures that the prediction, represented by h(t) Eq. (6) closely tracks the target (streamflow). However, given that h(t) is
determined as the product of the output gate o(t) and a function of the cell state c(t) (i.e., $h(t) = o(t) * \tan h(c(t))$), it is clear that
many different combinations of these trajectories can result in the same trajectory for h(t). So, given that the state variable to flux
relationship (output gate) is only weakly constrained in the current implementation of HydroLSTM, we should not expect to arrive
at a unique representation for the cell-state. To further constrain a HydroLSTM-based model to learn cell states (such as snow
water equivalent, water table depth, soil moisture, etc.) that align with hydrological understanding, we will necessarily have to add
extra information that regularizes the internal (latent space) behavior of the model. In ongoing work, we are exploring how the use
of predefined weight patterns (such as those that follow Gamma or Poisson distributional shapes), and/or training to multiple
catchments simultaneously while adding information regarding static catchment attributes might help to better constrain the learned
cell state trajectories.

Further, it is worth noting that while the HydroLSTM and LSTM representations have access to the same information sources,
they use that information in somewhat different ways. In the LSTM, the gates only have access to current time step information,
and a significantly larger number of cell-states is needed to obtain a given level of predictive performance. In other words, most




of the information about past system history that is relevant to making accurate predictions is encoded into the cell-states. In contrast, the HydroLSTM is provided access to much of that same historical information via the sequences of lagged input data that are fed into the gating mechanisms, and therefore information regarding the current "state" of the system can be encoded via a smaller number of cell states. Given that both architectures provide comparable predictive performance through a different
process of encoding the relevant information about the input-state-output dynamics of the catchment system, therefore, both representations can be considered to be valid.

## 8 Conclusions

We have proposed and tested a more interpretable LSTM architecture that better aligns with how we conceptualize the physical functioning of hydrologic systems. This gain in interpretability is achieved by modifying how the "state" of the system is tracked
(sequentially from the beginning to the end of a historical data set) and by providing the input, output, and forget gates with access to lagged sequences of historical data. We have named this modified architecture the HydroLSTM, to acknowledge the inspiration obtained from the isomorphic similarities of its cell-states to that of a hydrological reservoir unit.

The HydroLSTM architecture provides comparable performance to the original LSTM while requiring fewer cell states (as was demonstrated using data from 10 catchments from five hydroclimatically different regions). At the same time, the weights
associated with the sequences of lagged inputs of each gate display patterns (i.e. express characteristic features) that can help distinguish between catchments from different regions. A detailed examination of the impact of sequence length (a hyperparameter related to system memory time scales) indicates that this is an important architectural aspect that varies with location and can be at least partially associated with aridity. An additional degree of flexibility that should be incorporated into modeling frameworks would be the ability to learn this property from the data, so that a more truly globally applicable architecture can be achieved.  A
similar argument can be made for learning how many cell-states are needed while adhering to the principle of parsimony.

We propose that the sequenced patterns of the weights encode hydrological signature properties. This should be further investigated on a broader set of catchments on the set we used for this analysis. If the behavior we demonstrated here is found to be robust on larger sample sizes, this would open up a pathway to exploring how clustering based on such signatures can help to characterize catchments in terms of their similarities and differences, a task that has proven to be challenging (Singh et al., 2014; Ali et al.,
2012). We suspect that these weight patterns can eventually be regularized using fixed functional forms (e.g., by combining appropriate parametric basis functions) to reduce the number of parameters to be learned, and potentially further enhance hydrological interpretability by relating those parameters to catchment characteristics that are computable directly from data.

In conclusion, we have demonstrated that by looking under the hood of a machine learning representation with a view to making appropriate modifications to the architecture; it is possible to create ways to better extract useful information from the learning
process while retaining all (or at least most) of its strengths. That is an indication of how powerful our representations are, at the same time how constrained we could be if we do not understand them deeply. For that reason, it behooves us to choose those representations carefully and to be prepared to adapt and improve them in response to what we learn from our scientific explorations.




**Appendix A: Comparison between LSTM and Hydro LSTM for all the catchments.**

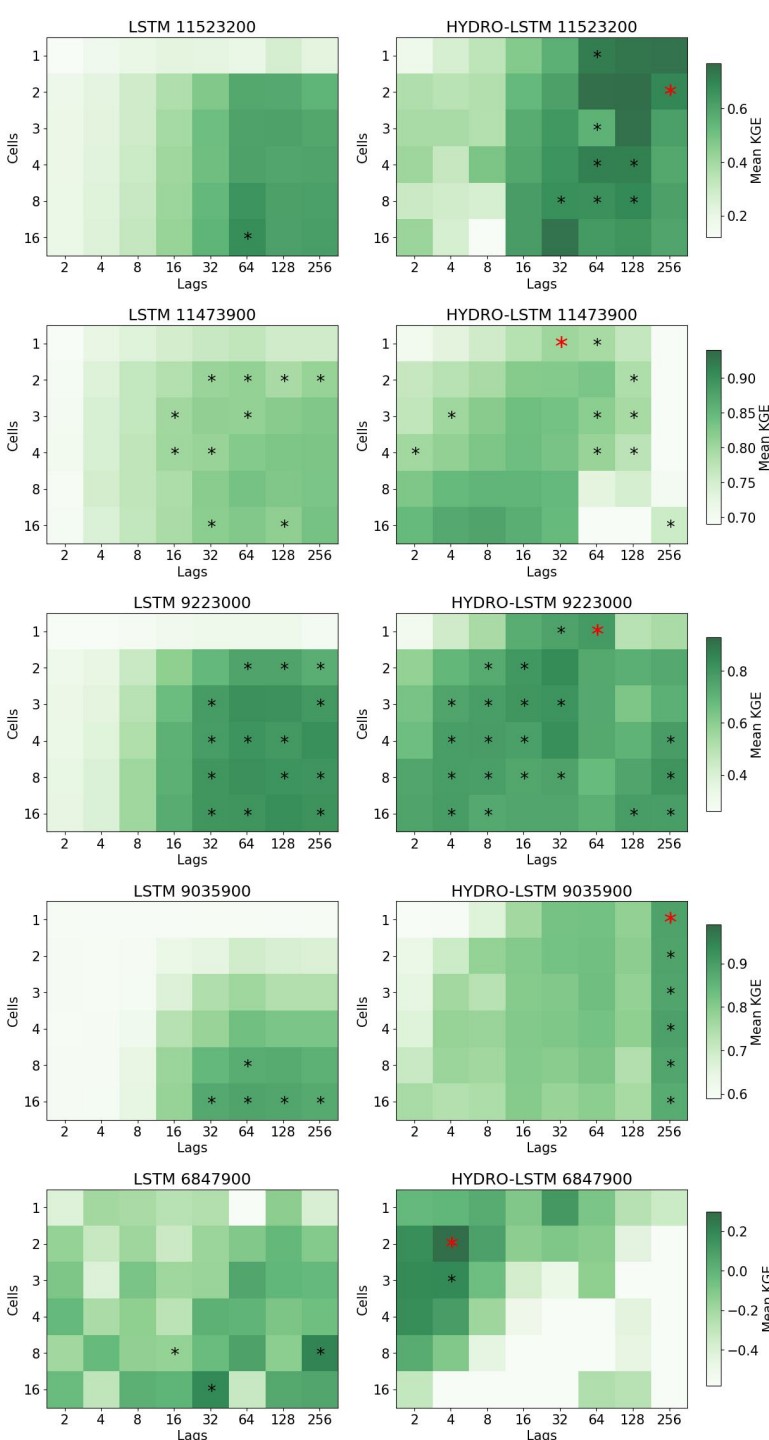

**Figure A1a**: Performance heatmap for different hyperparameter sets, number of cells, and lag days. Rows: Catchment studied.

Columns: representation. "*" shows the hyperparameter sets with no statistical difference in the mean with respect to the red "*".




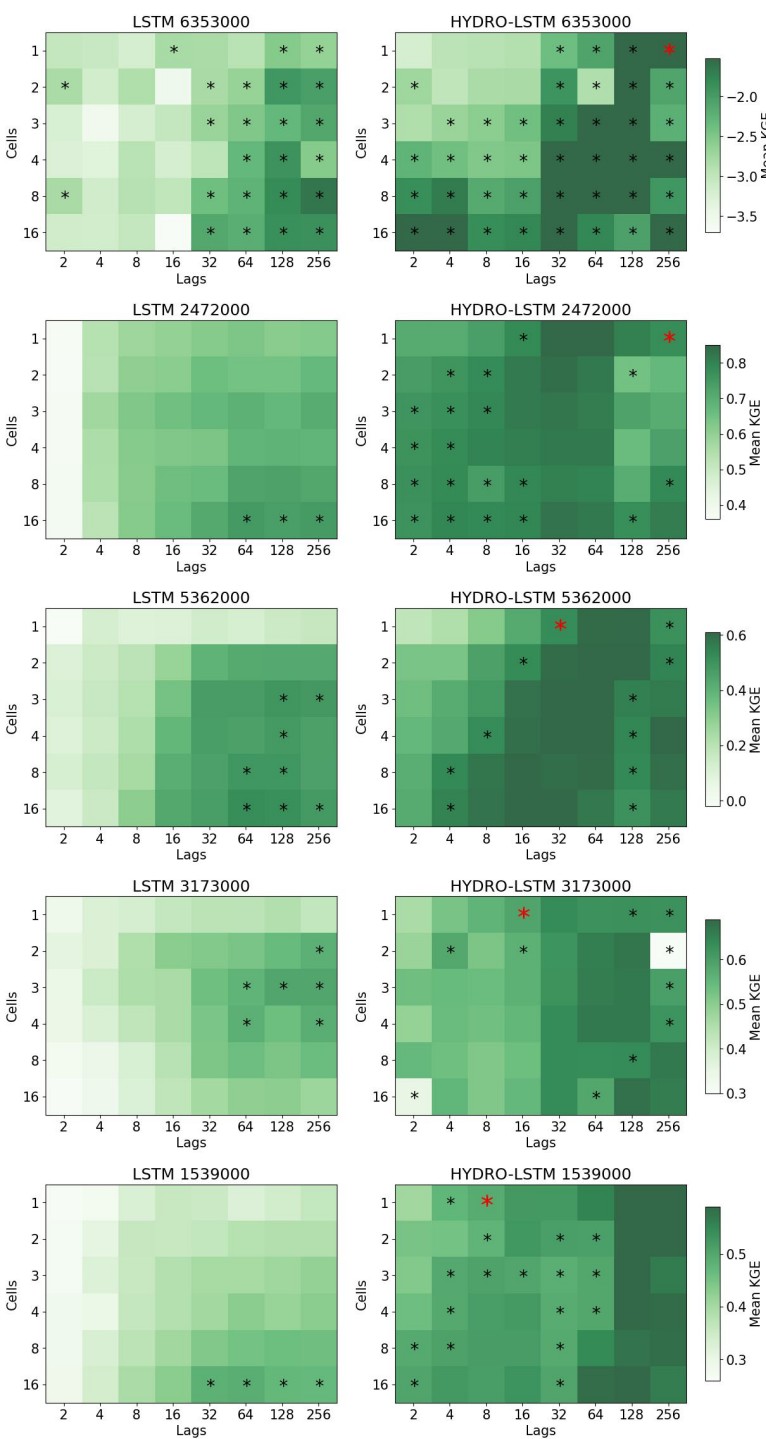

**Figure A1b**: Performance heatmap for different hyperparameter sets, number of cells, and lag days. Rows: Catchment studied. Columns: representation. "*" shows the hyperparameter sets with no statistical difference in the mean with respect to the red "*".





**Appendix B: Summary of the comparison between LSTM and HydroLSTM (first dataset).**

**Table B1. Comparison between LSTM and HydroLSTM.**

| Code | Name | Model | Parsimony | | Best performance |
|------|------|-------|-----|-------|------------------|
| | | | Lag | Cells | KGE |
| 11523200 | Trinity River above Coffee Creak, near Trinity Center, CA | HydroLSTM | 64 | 1 | 0.75 |
| | | LSTM | 64 | 16 | 0.67 |
| 11473900 | Middle Fork Eel River, near Dos Rios, CA | HydroLSTM | 32 | 1 | 0.88 |
| | | LSTM | 32 | 2 | 0.84 |
| 9223000 | Hams Fork below Pole Creek, near Frontier, WY | HydroLSTM | 32 | 1 | 0.84 |
| | | LSTM | 64 | 2 | 0.83 |
| 9035900 | South Fork of Williams Fork, near Leal, CO | HydroLSTM | 256 | 1 | 0.89 |
| | | LSTM | 64 | 8 | 0.89 |
| 6847900 | Prairie Dog Creek above Keith Sebelius Lake, KS | HydroLSTM | 4 | 2 | 0.28 |
| | | LSTM | 16 | 8 | 0.20 |
| 6353000 | Cedar Creek, near Raleigh, ND | HydroLSTM | 32 | 1 | -0.75 |
| | | LSTM | 16 | 1 | -1.62 |
| 2472000 | Leaf River, near Collins, MS | HydroLSTM | 16 | 1 | 0.87 |
| | | LSTM | 64 | 16 | 0.75 |
| 5362000 | Jump River at Sheldon, WI | HydroLSTM | 32 | 1 | 0.66 |
| | | LSTM | 128 | 3 | 0.51 |
| 3173000 | Walker Creek at Bane, VA | HydroLSTM | 16 | 1 | 0.68 |
| | | LSTM | 256 | 2 | 0.59 |
| 1539000 | Fishing Creek, near Bloomsburg, PA | HydroLSTM | 4 | 1 | 0.65 |
| | | LSTM | 32 | 16 | 0.49 |


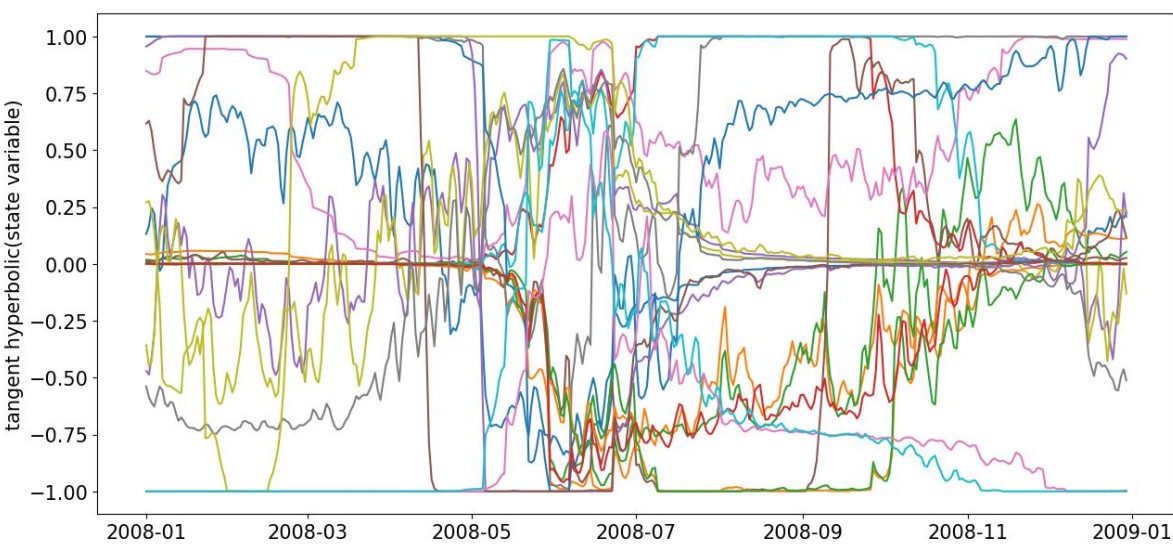

**Figure B1: Time evolution of the state variable across the ensemble of 20 models for catchment South Fork of Williams Fork, CO. It does not exist a unique evolution despite having good performance and one cell representation.**





**Figure C1a: Weight distribution in the three gates for HydroLSTM. Each row represents a different catchment. The confidence interval is the result of running 20 models with random initialization.**






**Figure C1b: Weight distribution in the three gates for HydroLSTM. Each row represents a different catchment. The confidence interval is the result of running 20 models with random initialization.**



**Code availability**

The codes to run the model and the Jupiter notebook used to create the figures are freely available at https://github.com/ldelafue/Hydro-LSTM.

**Data availability**

The CAMELS dataset is freely available from https://gdex.ucar.edu/dataset/camels.html.

**Author contribution**

De la Fuente, Ehsani, and Gupta participated in the initial conceptualization. De la Fuente and Ehsani (early stage) developed the formal analysis. Gupta and Condon participated in the methodology and supervision. De la Fuente developed the original draft and the entire team worked on reviewing and editing the document.

**Competing interests**

The authors declare no conflicts of interest relevant to this study.

**Acknowledgments**

This work was supported by Early Career Award NSF Grant 1945195 (Condon and De la Fuente), HydroGEN Project NSF Grant 2134892 (Condon, Gupta, and De la Fuente), and partially by the Chilean Government scholarship "Becas Chile, 2022" (De la Fuente). The authors would also like to acknowledge the feedback received from the CondonLab team and the reviewers.

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
