# Peer review of "Towards Interpretable LSTM-based Modelling of Hydrological Systems"

_EGUsphere, 2023_

## Community Comment (CC1)

**Comment on "Towards Interpretable LSTM-based Modelling of Hydrological Systems"**

**Summary of Paper:**

The objective of this paper is to allow for interpretability of the weights of an LSTM model for streamflow prediction. The authors approach this objective by making two changes to their baseline model (from Line 200 in the manuscript):

1) They use sequence-to-sequence prediction
2) They give each cell of the LSTM a sequence of lagged input data.

**Comment #1:**

I believe that there is a problem with one of the fundamental arguments that appears throughout the paper, related to the relationship between the model and the physical system.

- Line 10: *"Our architecture, called HydroLSTM, simulates behaviors inherent in a dynamic system, such as sequential updating of the Markovian storage"*
- Line 538: *"We have proposed and tested a more interpretable LSTM architecture that better aligns with how we conceptualize the physical functioning of hydrologic systems."*

What the authors did is actually break the Markovian nature of the LSTM. The regular LSTM is Markovian, in that the prediction at time t is dependent only on the state of the system, and the inputs at time t. By adding lagged inputs to each LSTM cell, the authors have made the model non-Markovian. This explicitly breaks the isomorphism between the LSTM and the physical system. The physical system is Markovian, since the watershed cannot "see" yesterday's rainfall, except through the effect on the various storage states within the system. The addition of lagged inputs actually aligns significantly *worse* with how we conceptualize physical hydrological systems.

Moreover, the authors claim (Line 45) that 45: *"It is possible, therefore, that either the corresponding LSTM-based models are not efficient (parsimonious) representations of the input-state-output dynamics, or that our conceptual hydrological models are overly simplified representations of reality (over-compression). In this paper, we make an argument for the former explanation."* I do not believe that this is a correct interpretation of the experiments in this paper. What the authors have done is to show that if you remove the Markovian assumption from the model, then you can replace information that in the physical system would be stored as a state

variable with (non-markovian) lagged input data, which the real system does not have access to (soil can't "see" rainfall from yesterday except through the current state of the watershed). I think you've just shown that non-Markovian models require less memory than Markovian models, which, in my opinion, is obvious.

It is the case that we do know that the LSTMs do not necessarily optimize to reduce the number of states. You can see this by doing an experiment where you emulate (exactly reproduce) a conceptual model - the LSTM typically requires more states than the "true" system (as defined by the conceptual model). This is a byproduct of inefficient (local) optimization, which is all that can be achieved using backpropagation. However, I don't think that the experiments in this paper demonstrate the claim, since you've allowed the number of states to be reduced due to breaking the memory requirement of the model - of course if you show the model yesterday's precipitation in today's state/output calculation, then the model doesn't need to remember as much information.

**Comment #2:**

I believe there is a misunderstanding about sequence-to-sequence (seq2seq) vs sequence-to-one (seq2one) prediction.

The authors state in line 150 *"In these areas, two primary assumptions are typically applied that may not hold in the dynamic environmental system: a) a finite relevant sequence length (finite memory time-scale), and the consequent possibility of 150 b) a non-informative system state initialization."*

They also state in line 200 *"the cell states are continually updated from the beginning to the end of the available dataset while maintaining the sequential ordering of the input drivers. This ensures that the cell states represent Markovian memories that are effectively of indeterminate length (as in traditional hydrological modeling, initialization is done only once at the beginning of the simulation period)"*

What the authors are describing is the difference between seq2seq vs. seq2one. The Kratzert papers use seq2one for *training*, but seq2seq for *prediction*. The reason that seq2seq is used for training is to help randomize the minibatch. Seq2seq allows for dividing the total training sample up so that any single minibatch contains samples from multiple different watersheds. This helps prevent spurious weight updates, which can significantly harm training. However, once the models are trained, they are applied as these authors describe — simulating the entire sequence without restarting (i.e., seq2seq prediction). The authors might notice that this is why the Krazert models require O(10) epochs for training while their seq2seq approach requires O(100) epochs for training.

It's also important to point out that this is not actually a difference in the model itself, but instead a difference in how the model is applied (seq2seq models are identical to seq2one models, just applied to different types of data). This is just a distinction in terms of how this is framed in the

discussion (the authors refer to this in line 199 as a difference in model structure, which it is not).

The authors should be aware that this seq2seq vs. seq2one distinction does not actually make a difference (other than the minibatch diversity thing during training, which is important). The reason why it does not matter is because of the forget gate. The forget gate can be thought of as a repeated multiplication operation. If the forget gates are not exactly one, then the memory of the model is limited by this repeated operation. Imagine repeatedly applying the forget gate operation on information in cell states from 300 time steps ago. If the forget gate has values close to one (i.e., fully open, meaning that they try to forget as little information as possible), then you are applying a repeated operation of multiplying by a number close to one several hundred times. As an example, $0.99 \char94 300 = 0.05$, meaning that even in a very optimistic case where the forget gate is almost completely open at all timesteps, only 5% of the information is left after 300 timesteps *purely because of numerical artifacts in the asymptotes of the forget gate activation functions.* It does not make very much sense to extend the input time sequence very much beyond this, simply because there will be no mathematical effect. This is why we are comfortable using seq2one for training, and the reason we use seq2seq for inference is because it's simply not necessary to do the extra computations necessary for seq2one when there is no minibatch.

I'd encourage the authors to test this – how much memory can the model have before it loses all sensitivity to inputs in the distant past? This can be answered using integrated gradients (we have done this experiment). Notice that if the authors wanted to avoid this numerical artifact, they could input a very long sequence directly into each cell state of the LSTM, however the number of weights in the model would explode. At some point, you might as well just use a time convolution (or similar), and not worry about the cell states at all.

**Comment #3:**

The authors cited my 2021 paper in the first paragraph of the introduction, however that paper does not say anything similar to what is claimed in the sentence where it is cited (and I strongly disagree with the opinion expressed in that sentence). I would kindly ask the authors to please either remove this incorrect citation or else modify it to more accurately reflect what is written in the paper they are citing.

**Comment #4:**

It would be very helpful if the authors would benchmark against an existing published paper. The authors have changed the set of basins that they train and test on (Line , as well as the train and test periods. This means that there is no way to know whether the authors have set up their LSTM in a way that matches the current state of the art. Their results are not directly comparable to anything that has previously been published. This should be easy, since the authors are using the same CAMELS dataset that numerous previous authors have used to

develop and test the baseline ML model used in this study. It is OK to make changes to the train and test settings, but it would be very helpful if the authors provided a community benchmark to help us know whether to trust their results. The authors might see [1] for and example where we felt that it was necessary to change the train/test periods for a specific experiment, but also published benchmarks against previous publications in the same paper, to ensure readers that our models were performing near state of the art.

**Comment #5:**

It is important to note that the integrated gradients method that the authors discuss in the introduction actually provides effectively the same information that this new method provides. Both methods give us relative importances on different model input channels. The authors could see [2] for an example of using integrated gradients to understand sensitivity of an LSTM to lagged input data. So what is this new method supposed to help us with? What can be learned from this that cannot be learned by using a method that does break the isomorphism between the model and the physical system (by making the model non-Markovian), and that does not harm the performance of the model?

After reading this paper, my main question for the authors is this: What problem are you trying to solve? What do you want to be able to do that can't already be done (and hasn't already been done)? Is there some limitation to what we can learn using explainability methods, and if so, what are those limitations?

**References:**

[1] Frame, J. M., Kratzert, F., Klotz, D., Gauch, M., Shalev, G., Gilon, O., Qualls, L. M., Gupta, H. V., and Nearing, G. S.: Deep learning rainfall–runoff predictions of extreme events, Hydrol. Earth Syst. Sci., 26, 3377–3392, https://doi.org/10.5194/hess-26-3377-2022, 2022.

[2] Kratzert, F., Klotz, D., Hochreiter, S., and Nearing, G. S.: A note on leveraging synergy in multiple meteorological data sets with deep learning for rainfall–runoff modeling, Hydrol. Earth Syst. Sci., 25, 2685–2703, https://doi.org/10.5194/hess-25-2685-2021, 2021.

---

## Author Comment (AC1)

**Response to Grey Nearing's comments.**

Firstly, we would like to thank you for reviewing and commenting on our paper. We find this discussion very interesting and feel that it will enrich the final version of the paper.

**Reviewer Comment #1:**

I believe that there is a problem with one of the fundamental arguments that appears throughout the paper, related to the relationship between the model and the physical system.

> ● Line 10: "Our architecture, called HydroLSTM, simulates behaviors inherent in a dynamic system, such as sequential updating of the Markovian storage"
> ● Line 538: "We have proposed and tested a more interpretable LSTM architecture that better aligns with how we conceptualize the physical functioning of hydrologic systems."

What the authors did is actually break the Markovian nature of the LSTM. The regular LSTM is Markovian, in that the prediction at time t is dependent only on the state of the system, and the inputs at time t. By adding lagged inputs to each LSTM cell, the authors have made the model non-Markovian. This explicitly breaks the isomorphism between the LSTM and the physical system. The physical system is Markovian, since the watershed cannot "see" yesterday's rainfall, except through the effect on the various storage states within the system. The addition of lagged inputs actually aligns significantly *worse* with how we conceptualize physical hydrological systems.

Moreover, the authors claim (Line 45) that 45: "*It is possible, therefore, that either the corresponding LSTM-based models are not efficient (parsimonious) representations of the input-state-output dynamics, or that our conceptual hydrological models are overly simplified representations of reality (over-compression). In this paper, we make an argument for the former explanation.*" I do not believe that this is a correct interpretation of the experiments in this paper. What the authors have done is to show that if you remove the Markovian assumption from the model, then you can replace information that in the physical system would be stored as a state variable with (non-markovian) lagged input data, which the real system does not have access to (soil can't "see" rainfall from yesterday except through the current state of the watershed). I think you've just shown that non-Markovian models require less memory than Markovian models, which, in my opinion, is obvious.

It is the case that we do know that the LSTMs do not necessarily optimize to reduce the number of states. You can see this by doing an experiment where you emulate (exactly reproduce) a conceptual model - the LSTM typically requires more states than the "true" system (as defined by the conceptual model). This is a byproduct of inefficient (local) optimization, which is all that can be achieved using backpropagation. However, I don't think that the experiments in this paper demonstrate the claim, since you've allowed the number of states to be reduced due to breaking the memory requirement of the model - of course if you show the model yesterday's precipitation in today's state/output calculation, then the model doesn't need to remember as much information.

**Author Response:**

Our interpretation of a Markovian process is that, for each time step, all of the information provided by the past data that is useful for making the next-time-steps prediction is contained in the system "states" (however those are determined). Based on this, one might argue that *neither* the LSTM nor the HydroLSTM is fully "Markovian".

The conventional LSTM implementation assumes some arbitrary initial value for the cell-states at some assumed/fixed lagged time in the past and then processes some past data sequentially (in a Markovian manner) to develop an estimate of important properties of the system state at time t (when a prediction for the next time step is desired). The assumption, therefore, is that there is no useful information (relevant to making that prediction) in the data representing times before that assumed

sequence length. Arguably, this goes against hydrological knowledge (whether correct or erroneous), which suggests that process memory can exceed 365 days, which is a very common sequence length used in LSTM-based hydrological modeling). Under such conditions, the conventional LSTM-based representation is not able to exploit all of the information that should otherwise be summarized in the representation of the system state for use in prediction, which arguably also breaks the Markovian assumption.

Similarly, the proposed HydroLSTM architecture also uses sequential updating "akin" to a Markovian process to estimate important properties of the system state at time t, for use in making the prediction for the next time step. And as with the LSTM, the gating functions define how the "cell-state" updating is done. However, because the gating functions use past lagged data to establish context for such gating, the gates have access to more information than is provided by the current forcing and cell-states. As you have pointed out, this also breaks the traditional Markovian nature of the cell-state representation and updating process, because it allows for important information regarding the overall "state of *the system*" to be explicitly provided via sequence lagged data fed to the gating functions to be used in determining how open or closed they are. In other words, the "cell-states" now do not represent the entire "*state*" of the system because they no longer contain *all* of the information provided by the past data that is useful for making the next-time-steps prediction. Arguably the true "system state" is now represented by a combination of the "*cell-states*" *and* the sequence information provided to the gates, since both are used to inform the next time-steps prediction.

Which way of breaking the classical "Markovian" behavior is better or worse is arguably dependent on our goal. And it is not clear that cell-state summaries determined primarily from "Markovian-like" processing of the mass and energy-related time-series provided as inputs to the model are necessarily sufficient to characterize the true "state" of the system at any point in time. Therefore, our view is that whether or not a classical "Markovian" or "non-Markovian" (or even "partially Markovian") representation is useful for gaining insight into the internal relationships learned by a Machine Learning model is not the most important thing.

Regarding the comment that the LSTM representation does not necessarily optimize to reduce the number of states, this is an interesting issue that would be useful to develop a better understanding of. It could also, perhaps, be a consequence of the limitations imposed by the specific architectural choices available to the LSTM (including, for example, the forms of the activation functions used). And, as you mention in the next comments, it could also be associated with the non-conservative behavior of the forget gate. We are of the opinion that more investigation should be done to explore improvements or modifications to the current LSTM representation, at least when applied in a hydrological context.

In summary, we thank you for raising the discussion about the Markovian nature of the process, as it is a topic that will enrich the paper – as a consequence we plan to add a small discussion to the revision. Moreover, to avoid misunderstandings with the concept of Markovian we will use "quotes" when using this concept in reference to the proposed HydroLSTM architecture.

**Reviewer Comment #2:**

I believe there is a misunderstanding about sequence-to-sequence (seq2seq) vs sequence-to-one (seq2one) prediction.

The authors state in line 150 "*In these areas, two primary assumptions are typically applied that may not hold in the dynamic environmental system: a) a finite relevant sequence length (finite memory time-scale), and the consequent possibility of 150 b) a non-informative system state initialization.*"

They also state in line 200 "*the cell states are continually updated from the beginning to the end of the available dataset while maintaining the sequential ordering of the input drivers. This ensures that the cell states represent*

*Markovian memories that are effectively of indeterminate length (as in traditional hydrological modeling, initialization is done only once at the beginning of the simulation period)"*

What the authors are describing is the difference between seq2seq vs. seq2one. The Kratzert papers use seq2one for *training*, but seq2seq for *prediction*. The reason that seq2seq is used for training is to help randomize the minibatch. Seq2seq allows for dividing the total training sample up so that any single minibatch contains samples from multiple different watersheds. This helps prevent spurious weight updates, which can significantly harm training. However, once the models are trained, they are applied as these authors describe — simulating the entire sequence without restarting (i.e., seq2seq prediction). The authors might notice that this is why the Krazert models require O(10) epochs for training while their seq2seq approach requires O(100) epochs for training.

It's also important to point out that this is not actually a difference in the model itself, but instead a difference in how the model is applied (seq2seq models are identical to seq2one models, just applied to different types of data). This is just a distinction in terms of how this is framed in the discussion (the authors refer to this in line 199 as a difference in model structure, which it is not).

The authors should be aware that this seq2seq vs. seq2one distinction does not actually make a difference (other than the minibatch diversity thing during training, which is important). The reason why it does not matter is because of the forget gate. The forget gate can be thought of as a repeated multiplication operation. If the forget gates are not exactly one, then the memory of the model is limited by this repeated operation. Imagine repeatedly applying the forget gate operation on information in cell states from 300 time steps ago. If the forget gate has values close to one (i.e., fully open, meaning that they try to forget as little information as possible), then you are applying a repeated operation of multiplying by a number close to one several hundred times. As an example, $0.99 \char`^ 300 = 0.05$, meaning that even in a very optimistic case where the forget gate is almost completely open at all timesteps, only 5% of the information is left after 300 timesteps *purely because of numerical artifacts in the asymptotes of the forget gate activation functions.* It does not make very much sense to extend the input time sequence very much beyond this, simply because there will be no mathematical effect. This is why we are comfortable using seq2one for training, and the reason we use seq2seq for inference is because it's simply not necessary to do the extra computations necessary for seq2one when there is no minibatch.

I'd encourage the authors to test this – how much memory can the model have before it loses all sensitivity to inputs in the distant past? This can be answered using integrated gradients (we have done this experiment). Notice that if the authors wanted to avoid this numerical artifact, they could input a very long sequence directly into each cell state of the LSTM, however the number of weights in the model would explode. At some point, you might as well just use a time convolution (or similar), and not worry about the cell states at all.

**Author Response:**

We agreed that your description of seq2seq and seq2one sounds like what we described. However, we are trying to describe something slightly different. In hydrological models, a warm-up period is commonly used to establish the initial value of the state variable. After this period, the state variable is updated with each new piece of information (in our case, daily precipitation, and temperature). The warm-up period can be thought of as a seq2one process, while the posterior updating of the state resembles a seq2seq process. However, during the training of a seq2one model, this warming-up process is repeated for each time value in the time series, which can result in the model learning an average sequence length that does not consider anomalies with respect to the mean. To address this issue, the state value could be fed at the beginning of each training period, but this approach would not allow for the randomization of samples described in the comment. The process of beginning from zero constantly to define the state variable is what we describe as a process that is not like what we do when we solve a dynamic system.

The mention of seq2seq in the paper was to try and express that more ideas can be built up over LSTM than to mention it as another architecture. We will clarify this in the next version of the paper.

As you mentioned very well, the forget gate is non-conservative (asymptotic to 1) which truncates the past information until it is insensitive. Therefore, new mechanisms should be explored to deal with that. The idea of using time convolution is well-taken because HydroLSTM is basically using a time convolution inside of the gates. What we have found is that such filters can be informative, in an interesting way, about the hydrological relationships built inside the representation, which is the final goal of the paper.

**Reviewer Comment #3:**

The authors cited my 2021 paper in the first paragraph of the introduction, however that paper does not say anything similar to what is claimed in the sentence where it is cited (and I strongly disagree with the opinion expressed in that sentence). I would kindly ask the authors to please either remove this incorrect citation or else modify it to more accurately reflect what is written in the paper they are citing.

**Author Response:**

We apologize for this inaccuracy. The reference will be deleted in the revised version.

**Reviewer Comment #4:**

It would be very helpful if the authors would benchmark against an existing published paper. The authors have changed the set of basins that they train and test on (Line , as well as the train and test periods. This means that there is no way to know whether the authors have set up their LSTM in a way that matches the current state of the art. Their results are not directly comparable to anything that has previously been published. This should be easy, since the authors are using the same CAMELS dataset that numerous previous authors have used to develop and test the baseline ML model used in this study. It is OK to make changes to the train and test settings, but it would be very helpful if the authors provided a community benchmark to help us know whether to trust their results. The authors might see [1] for and example where we felt that it was necessary to change the train/test periods for a specific experiment, but also published benchmarks against previous publications in the same paper, to ensure readers that our models were performing near state of the art.

**Author Response:**

We are not sure what you mean by changing the basin of training and testing, which probably means we should improve the description of our experiments in the next version.

We conducted two experiments. In the first experiment, we compared both representations using only 10 catchments, training one model per catchment. Each catchment was calibrated using data from January 1, 1980 to September 30, 2000. We then used the next four years to select the best epoch and the final period to present the results.

From the results of this experiment, we found that a single HydroLSTM cell had reasonably "good" performance compared to the best possible configuration of lag data and the number of cells. Therefore, in the second experiment, we explored what we could learn from this simplified representation. We independently trained one HydroLSTM cell for each of the 588 selected catchments (including the previous 10), keeping the same procedure as in the first experiment (i.e., splitting the data).

Please note, as was mentioned in the paper, that our goal is _not_ to demonstrate that HydroLSTM has a better performance than the LSTM. It is highly probable that state of the art in LSTM representations and those that use of more cells than we used, could beat HydroLSTM in terms of performance. Instead, our aim was to explore the possibility of parsimonious representations (in terms of numbers of cell-states) with the goal of gaining insights into the interpretability of parameters and state

variables. Given this purpose, we thought it was most important to select a subset of basins that represent the range of hydrologic behaviors we want to capture, rather than selecting basins where we could compare to previous studies.

**Reviewer Comment #5:**

It is important to note that the integrated gradients method that the authors discuss in the introduction actually provides effectively the same information that this new method provides. Both methods give us relative importances on different model input channels. The authors could see [2] for an example of using integrated gradients to understand sensitivity of an LSTM to lagged input data. So what is this new method supposed to help us with? What can be learned from this that cannot be learned by using a method that does break the isomorphism between the model and the physical system (by making the model non-Markovian), and that does not harm the performance of the model?

After reading this paper, my main question for the authors is this: What problem are you trying to solve? What do you want to be able to do that can't already be done (and hasn't already been done)? Is there some limitation to what we can learn using explainability methods, and if so, what are those limitations?

**Response:**

We agree that the gradient method effectively provides the same information. However, the HydroLSTM uses that information explicitly in the gates. This allows the user to more easily understand and visually interpret the "feature importance" encoded in the model. For that reason, the title of our paper is focused on interpretability, rather than on presenting HydroLSTM as any kind of new state of the art. Given that some people are still reluctant to use machine learning methods, because they are seeking more than just predictive performance, we feel that finding ways to make machine learning methods as interpretable as possible is a valuable goal.

---

## Author Comment (AC2)

Dear Tadd,

Thank you for your kind comments. We will incorporate your suggestions and clarify parts that were not clear enough.

Best regards,

De la Fuente et al.
* * *
Hello,

Thank you for the lovely preprint. I enjoyed reading your work and offer the following suggestions below. I believe the paper should be reconsidered for HESS, with major revisions, and look forward to reading the next submission.

Best,

Tadd Bindas
* * *
Editorial questions:

1.  Does the paper address relevant scientific questions within the scope of HESS?
    1.  Yes.
2.  Does the paper present novel concepts, ideas, tools, or data?
    1.  The concept proposed by their HydroLSTM model is novel. The authors are looking to add more interpretability to the LSTM architecture and get similar results with fewer cell-states using the HydroLSTM code they developed.
3.  Are substantial conclusions reached?
    1.  I'm not sure. As a summary of my understanding of the paper: the results obtained from their first experiment showcase that a simplified LSTM framework (similar to our understanding of a reservoir) can use one cell to learn a relationship between inputted forcings. The second experiment shows how their model performs when compared to 588 CAMELS basin observations.

        **Response**: We agree. We would like to add that the second experiment explores how lag memory is another hydrological characteristic that is encoded in the weight pattern. This finding reinforces the idea that catchment attributes play a role distinct from meteorological forcing.

2. My confusion arises with how the authors train their HydroLSTM and LSTM in experiments 5 and 6. From what I've read, and understood from talks at conferences, LSTM models should be trained using all basin data, then tested at individual sites using either a PUB, PUR approach, or median NSE/KGE metric for all catchments. I do not believe the authors are doing this, thus, I am curious if training their HydroLSTM and LSTM models on all catchments would show the same results.

   **Response**: The approach mentioned is commonly used when the goal is to demonstrate the temporal or spatial consistency of a global-scale model. In our case, we are focused on predicting at a single catchment scale and aiming to determine the minimum complexity required to achieve similar performance, adhering to the principle of parsimony. Our findings indicate that incorporating contextual information in the gates simplifies the number of cell states and allows the weight patterns to encode hydrological knowledge. However, it is important to note that the behavior of total lag and weight patterns is specific to each catchment. Therefore, it becomes necessary to introduce appropriate regularization techniques to enable knowledge transfer between catchments. We are currently studying this step and plan to incorporate it in future applications of a global HydroLSTM representation.

4. Are the scientific methods and assumptions valid and clearly outlined?
   1. Yes. Table 1 does a good job of showing similarities between Storage and LSTM equations.

5. Are the results sufficient to support the interpretations and conclusions?
   1. I believe more work needs to be done to validate the conclusion that HydroLSTM provides comparable performance with LSTM, but with added interpretability. A PUR or PUB experiment to see how a HydroLSTM trained on all CAMELS basins performs would be appreciated.

      **Response**: Adding a comparison with LSTM in terms of PUR and PUB would indeed provide valuable information if the sole purpose were to evaluate performance. However, such a step would require further research into the regionalization of weights within the gates, which is currently under development. Therefore, we will modify the text to explicitly state that our conclusions are only applicable at a single catchment scale.

6. Is the description of experiments and calculations sufficiently complete and precise to allow their reproduction by fellow scientists (traceability of results)?
   1. Almost. I still need clarification on the model training procedure.

> **Response**: We will incorporate more details about the training procedure and provide the reasons for adopting this approach.

7. Do the authors give proper credit to related work and clearly indicate their own new/original contribution?

    1. Yes

8. Does the title clearly reflect the contents of the paper?

    1. Yes

9. Does the abstract provide a concise and complete summary?

    1. Yes

10. Is the overall presentation well-structured and clear?

    1. Yes

11. Is the language fluent and precise?

    1. Yes

12. Are mathematical formulae, symbols, abbreviations, and units correctly defined and used?

    1. It would be appreciated to italicize all equations when in-line. It was hard to read/locate them amongst the text. There are also some repeated variable names (See the comments for an example).

       **Response**: The suggestion will be incorporated in the next version of the paper.

13. Should any parts of the paper (text, formulae, figures, tables) be clarified, reduced, combined, or eliminated?

    1. The model training could be a little clearer (Similar to the above comment).

       **Response**: We will include additional details about the training procedure and elaborate on the reasons for adopting this specific approach in the next version of the paper.

14. Are the number and quality of references appropriate?

    1. Yes

15. Is the amount and quality of supplementary material appropriate?

    1. Yes

Major Comments:

- Can you italicize all in-line variables and equations? It's hard to determine which parts of the text describe equations/LSTM properties. In some cases, I've had to reread a paragraph multiple times to search for an equation I missed.

  **Response**: The suggestion will be incorporated in the next version of the paper.

- (Lines 245) Are any static attributes used in model training?

  **Response**: Static attributes are not utilized in the current approach since each catchment is trained locally. Including static attributes would essentially introduce a biased term that is already accounted for in each gate. Hence, for the sake of consistency and avoiding redundancy, static attributes are not incorporated into the training process.

- (Lines 252-259) I suggest swapping Calibration, Selection, and Evaluation periods with the training, validation, and testing periods within the parentheses. It looks like you are using the train, validation, and test verbiage throughout the paper, and only referring to calibration, selection, and evaluation periods once (Line 427) after being defined.

  **Response**: We will review the paper using only the terminology of Calibration, Selection, and Evaluation.

- (Section 5.1) Would it be possible to include a PUR analysis rather than a 10-basin (PUB) holdout? So, rather than having two basins from each region, you would test on all gages within a snowmelt-dominant or Recent rainfall-dominant region. I believe this study would benefit from comparing how each LSTM performs on regions not included in the training set. This analysis would strengthen the claim that HydroLSTM has similar model performance to LSTM, but with heightened interpretability.

  **Response**: We acknowledge the value of PUR and PUB analysis in evaluating model performance. However, it is important to note that the current HydroLSTM representation is not designed to handle multiple catchments simultaneously. This limitation arises from the weight pattern within the gates. The lag and maximum weight values of one catchment cannot be directly transferred to another catchment solely through catchment attributes, as the attributes need to modify the weight pattern. Therefore, in order to transfer the knowledge learned from one catchment to another with similar catchment attributes, appropriate regularization techniques (such as regionalization) must be incorporated into the HydroLSTM architecture.

- (Line 310) How many total catchments were included in the training period? It is mentioned in Section 6.1, but not in 5.1. Is it just one catchment?

**Response**: Line 310 and Table 2 describes the 10 catchments used in the calibration (training) period. Experiment 1 (section 5) and Experiment 2 (section 6) are calibrated using one model per catchment but with a different total number of catchments in the analysis (10 and 588 catchments respectively).

- (Line 428) From my understanding of the literature, the best-performing LSTM models are using forcings, and attributes, from all basins in their inputs. For example, if there are 588 catchments, all catchments would be included in the training set. Then, testing would be done on all catchments, to determine a median KGE. Is training HydroLSTM on all catchments, or using basin attributes, something you have explored? Is the optimal lag memory hyperparameter the reason against having an entire CAMELS-trained LSTM? More explanation would be appreciated.

  **Response**: The method described is commonly used to assess the performance of different architectures, aiming to identify the best-performing option. However, in our study, our focus is primarily on the interpretability of what is learned by the architecture at each step. As a result, the analysis conducted at a single-level catchment can be considered an initial step before proceeding to train at a global scale.

  Based on this analysis, we have reached the conclusion that important hydrological properties, such as total lag memory and weight patterns, are encoded in the representation. This behavior is desirable for achieving interpretability. However, it also highlights a structural distinction in how meteorological forcing and catchment attributes should be processed.

  Due to this observed difference, we have not presented a global model in this particular paper.

- (Section 6) Is it possible to add a comparison against an LSTM applied to a large sample of catchments?

  **Response**: Given the current limitations of HydroLSTM, which can only be tested at a single catchment scale, while LSTM has already been tested at a global scale, a direct comparison between the two representations may not yield informative conclusions. However, we are currently working on incorporating a specific regularization technique that would enable the comparison of two global models.

- (Section 6) Is it possible to add a PUB comparison to this section?

  **Response**: As mentioned in the previous comments, a comparison under the current conditions is not possible.

Minor Comments:

- (Affiliations) The s in the United States is cut off

  **Response**: We will correct that mistype.

- (Line 58) Is Expected Gradient supposed to be capitalized?

  **Response**: The capitalization will be modified.

- (Line 107, Line 120) The symbol for the output gate, and the time-constant value, are both o. This could lead to some confusion.

  **Response**: We use the same letter to represent that both have the same behavior however we agree on extra differentiation should be added to avoid confusion.

- (Line 130-135) Physical state and informational state don't need to be italicized.

  **Response**: Those words were italicized because we considered them a simplified (even colloquial) characterization of the state. We will evaluate to change them in terms of the comments of other reviewers.

- (Table 1) Are the brackets supposed to be facing outward? (ex: $o = ]0,1[$)

  **Response**: We used outward brackets to emphasize the asymptotic behavior of sigmoid and hyperbolic tangent functions. In the case of the linear reservoir, we can use inward brackets to show that they can take on zero or one value, despite being extreme cases of the linear reservoir.

- (Line 212) Typo. There needs to be a space inside Wand

  **Response**: We will fix this typo.

- (Line 253) I believe you mean "Commonly referred to as Training, Validation, and Testing." You used evaluation twice in this part.

  **Response**: We will fix this typo

- (Line 286) You didn't establish what a testing period is (see earlier comment for Line 253). Testing should be replaced with "Evaluation."

  **Response**: We will fix this typo

- (Line 304) The header "5 Experiment 1" reads weird. Maybe change to "5 First Experiment?"

  **Response**: We will replace the title with "First Experiment"

- (Figure 3) It may be clearer to the reader that rows are the Catchment Studied if you put the gage number on the row's y-axis in bold above "Cells."

  **Response**: We will modify Figure 3 following the suggestion.

- (Line 417) Same as the above comment. Maybe replace this with 6 Second Experiment. The section title reads weird.

  **Response**: We will replace the title with "Second Experiment"

- (Line 439) There is an unnecessary space before "However"

  **Response**: We will fix this typo.

---

## Author Comment (AC3)

Major Comments:

1. The structure of the manuscript may need to be optimized to make it easier to read. For instance, Section 2.3 is titled "Differences with the Hydrologic Reservoir", but I am hard to get the differences between LSTM and the Hydrologic Reservoir; some terms, such as optimal lag memory, significant values for weights and so on, are not well defined in the manuscript; I am confused about how Experiment 2 helped with the topic.

   **Response**: First, we describe the similarities between both representations and the slight differences within them. In section 2.3, we outline what we consider the main two differences: state evolution and gating behavior. The first one describes how LSTM undergoes continuous warming up at each time step to approximate the state value. The second one describes the information used to infer the gating behavior. Since section 2.3 is fundamental for understanding the proposed structure, we will add more details and clarifications in this section.

2. The manuscript uses a long length to compare linear reservoir model and LSTM. However, the linear reservoir model is not used in the case study. Is it possible to use the linear reservoir model as the benchmark model to relate the parameters of the linear reservoir model and LSTM in order to discuss the physical meaning of LSTM parameters.

   **Response**: The analogy between the linear reservoir and LSTM is employed solely to elucidate the functioning of the representation. The concept of utilizing a conceptual lumped model as a benchmark in terms of performance and certain global behaviors has been employed in prior research (De la Fuente et al., 2023). Nevertheless, this comparison serves only to comprehend the advantages and shortcomings of each representation. Extracting the knowledge encoded within an ML model necessitates additional steps, which are currently being explored in the paper.

   The relationship between the parameters used by a linear reservoir and LSTM has not been explored. However, the differences between the two representations in terms of states (water vs. information), state tracking (continuous vs. warming up), and gate behavior (constant vs. dynamic) do not guarantee a correct and unique relationship between them.

3. My primary concern is that the HydroLSTM/LSTM is not well validated. From Figure 4, the performance of HydroLSTM/LSTM is not very good. Only 5 out of 10 basins have a KGE value larger than 0.7. In some previous studies, most runoff simulations with a local LSTM can obtain a KGE greater than 0.7. I am concerned about whether

the parameters and structure of the ML model are well set. Further, for a ML model with poor performance, the interpretability of the model does not seem to be very meaningful. Also, I wonder why the authors did not use more catchments to verify the reliability of the model. In my opinion, the summary of 588 catchments makes Figure 4 more credible.

**Response**: Local models have the potential to achieve overall good performance when the inputs are carefully selected. However, in our case, we are utilizing a parsimonious representation that only includes precipitation and temperature as inputs. This limitation restricts the maximum performance regardless of where the representation is applied.

In Figure 4, our objective was to demonstrate that under these simplistic conditions, LSTM and HydroLSTM exhibit similar performance. Furthermore, the selection of catchments in our study encompasses representative regions with traditionally good performance (wet catchments) and poor performance (arid catchments), which is consistent with the range of results shown in Figure 4b.

Additional cases could be included in Figure 4; however, the overall situation is unlikely to change significantly since HydroLSTM is merely a modification of the LSTM representation. Therefore, in terms of performance, they are expected to perform similarly, resulting in more data points clustered around the 1:1 line. Another limitation is the computational time required, as each catchment involves 20 runs multiplied by 8 lags (2, 4, 8, 16, 32, 64, 128, and 256 days) multiplied by 6 cells (1, 2, 3, 4, 8, 16 cells) multiplied by 2 representations, resulting in a total of 1920 models. This computational constraint restricts the analysis to a subset such as the entire dataset.

4. From Figure 5, the uncertainty of the model parameters seems large. The large uncertainty of parameters may make the model less interpretable. I think it is necessary to explain the effect of parameter uncertainty on the model.

   **Response**: We acknowledge that the uncertainty in the weight values is substantial, indicating a high degree of freedom and equifinality issues. However, the overall pattern per catchment remains consistent (Fig. C1b), which is a novel finding. This is noteworthy because weight values in machine learning models are typically considered random and non-interpretable. We consider this as one of the key outcomes of our study since it signifies the potential for extracting knowledge and enhancing interpretability from ML models. We will provide further explanations regarding the uncertainty in our upcoming revisions, adding more clarity to this aspect.

5. Why does Figure 5 use the logarithmic horizontal axis? If using a regular coordinate axis, I think it is hard to distinguish the fluctuation of precip weights between 0-10 days and after 10 days in the ID11473900 catchment. The fluctuation of Pot. Evapot. Weights in a regular coordinate axis seem to be a periodic variation in the ID9035900 catchment. The logarithmic horizontal axis may mislead readers into thinking that there is a trend from 0-1 days.

    **Response**: We opted for a logarithmic scale because the highest weight values are typically found within the 0-10 day range (Figure C1), and the relative importance of past information tends to diminish for longer lags. However, we observed some periodic behavior for longer lags, indicating that weight distributions carry informative signals about the relationship utilized by the ML model, and these distributions are specific to each catchment. To prevent any potential misinterpretation, we will include figures on a regular scale in the supplementary material, along with further explanations regarding the use of a logarithmic scale.

6. The manuscript analyses the physical meaning of the output gate. I'm wondering if the forget gate and input gate have a corresponding interpretation.

    **Response**: Effectively, the forget and input gates do possess some level of interpretability. However, their interpretation is more closely tied to the state variable and the nature of the input employed. Consequently, interpreting these gates directly is not feasible without appropriate regularization. For instance, the state variable may be storing diverse forms of pertinent information, making it difficult to determine the exact extent to which the model should remember or forget. Currently, there are ongoing works that concentrate on imposing constraints on the storage of specific entities such as volume and energy. In such cases, it may be possible to derive meaningful interpretations.

7. Why does Experiment 2 classify the catchments with Aridity rather than catchment dominance factors in Experiment 1?

    **Response**: The criteria presented in Table 1 were indeed developed using 160 catchments of the MOPEX dataset. Since this research used 588 catchments of CAMELS dataset, with only a small overlap between them, a direct comparison between the two datasets is not feasible. However, to address this challenge, we are exploring the possibility of incorporating clustering techniques for the catchments under study. By employing clustering, we aim to establish a meaningful comparison between the evaluated catchments and the criteria presented in Table 1, despite the differences between the datasets.

8.  How is the optimal number of lagged days obtained in Experiment 2? From Figure 8, I think the optimal numbers of lagged days of the catchments with AI<0.6 and AI>1.0 are also 128 days. There needs to be more discussion about the relationship between required memory time scales and aridity.

    **Response**: The description of the best lag is based on the median value (represented by the red line in the boxplot). However, it is important to note that this value serves as an overall summary of the trend where increasing lag corresponds to higher aridity levels. It is crucial to recognize that catchment memory is influenced by various factors beyond just aridity. Therefore, it is not possible to strictly define the "best" lag for a specific level of aridity. Rather, we can only observe that a relationship exists between aridity and lag. We will provide additional discussion on this topic to further elaborate on the complexities and limitations involved.

Minor Comments:

1.  We usually use the Hydrologic Reservoir model rather than the "Hydrologic Reservoir". Just a suggestion.

    **Response**: We will incorporate the suggestion.

2.  Table 2. It is necessary to explain the difference between "Recent" and "Historical".

    **Response**: We will include a summary in terms of the classification mentioned in the paper.

3.  Figure 4. How to choose the "red *"?

    **Response**: The "red*" indicates the best performance achieved by Hydro-LSTM when using one or two cells, where at least one "*" is present in the LSTM parameter sets. We employ this approach to demonstrate that Hydro-LSTM exhibits a parsimonious state representation while achieving similar performance compared to LSTM.

De la Fuente, L. A., Gupta, H. V., & Condon, L. E. (2023). Toward a multi-representational approach to prediction and understanding, in support of discovery in hydrology. *Water Resources Research*, 59, e2021WR031548. https://doi.org/10.1029/2021WR031548

---

## Author Comment (AC4)

**Egusphere-2023-666 "Towards Interpretable LSTM-based Modelling of Hydrological Systems" LA. De la Fuente, MR. Ehsani, HV. Gupta and LE. Condon**

This article is really well written and constructed, making it an easy manuscript to read. I find little to critique about the presentation, but will talk in generalities from a hydrological modellers perspective.

> **Response**: Thanks for your compliment. We appreciate your contribution and comment.

Temporal weighting of lagged events is highly reminiscent of unit hydrograph theory and is potentially another interpretation of the weights obtained (e.g., Sherman 1932; Lienhard 1964; Rodriguez-Iturbe and Valdes 1979). Similarly, the number of cell states may be broadly associated with the number of linear reservoirs in series that produce these unit hydrographs (e.g., Ocak and Bayazit 2003). Clearly a unit hydrograph that partitions daily total runoff into an hourly signal (for estimating peak flow for example) is not the same, but the concepts are equivalent when representing inputs as a temporal output distributed with the memory of prior inputs.

> **Response**: We agree. The unit hydrograph and its convolution effect evoke many aspects of what we found in the weight distribution. This similarity helps differentiate the short-term effect of the forcing (unit hydrograph behavior) without a state variable, from the long-term effect produced by the water stored in the catchment (baseflow). As a result of both effects, a non-linear behavior emerges in the input-output response of the streamflow. This explanation will help us to understand why these patterns emerged in the gates. We will add this additional explanation about the interpretability of the weights.

The titles of Sections 5 and 6 should be more descriptive than "Experiment 1" and "Experiment 2", maybe "Comparison of LSTM and HydroLSTM across hydrological regime" and "HydroLSTM performance with a single cell state".

> **Response**: We will make the titles of sections 5 and 6 more specific to their respective topics.

These suggestions are very rough, but the opening paragraph of each section should follow logically from the section title and expand upon it. It is unclear how the 10 catchments in Section 5 were selected, but it interesting to observe that in each of the five hydro-climate regimes that LSTM had one low cell state result, and one much higher. Was this deliberate or simply to support the later message regarding spatial variability

of lag, that definitive patterns of number of cell states or lag are difficult to establish based on hydro-climate for multiple cell state representations?

> **Response**: We will enhance the introduction of this section. Regarding the selection of the catchments, they were chosen randomly from each of the regions defined in the reference study (Jiang et al.,2022). This random selection illustrates the difficulty of determining the optimal lag solely based on performance metrics. It further emphasizes the necessity of additional regularization to draw more robust conclusions about hydrological features.

Please be careful in your equations that the hyperbolic tangent function "tanh" stays a single word and not split to "tan$_h$h" with a space, as I notice the language of tangent hyperbolic in the text.

> **Response**: We will check the correct writing of the tangent hyperbolic.

Figure 6a has four clearly inferior lag times (4, 8, 16 and 32 days) with the other three (64, 128 and 256 days) being the same for all practical purposes for KGE>0.4. It is hard to reconcile in the text (S6.2) that the graph has "saturated" at lag=256 if very similar results are obtained with lag=64 and 128, and there are no lag values >256 to confirm it.

> **Response**: We agree. The saturation region is between 64 and 256 days for catchments with KGE > 0.4. For KGE < 0.4, the curve of 256 days is the best option, which is why this one is used in the text. However, the main point of this figure is to demonstrate that a fixed memory might not be the optimal choice when dealing with a large number of catchments, considering the local dependency of that hyperparameter.

With Figure 6b it is not surprising that modelling with a single cell state is more difficult with increased aridity, as this is well known in the standard hydrologic modelling literature (e.g., Pilgrim et al. 1988).

> **Response**: We agree; however, this behavior is present in lumped and ML models, as explored previously (De la Fuente et al., 2023), suggesting that the issues are not in the architecture used. For that reason, presenting this figure should reinforce that idea. We will add some comments about that.

It also seems that the general success of HydroLSTM with a single cell state alludes to the usefulness (success?) of simple lumped hydrologic models such as GR4J or SIMHYD with few parameters that may be physically interpretable.

> **Response**: This is the inspiration for the paper: a parsimonious representation should be preferred if additional complexity cannot improve performance.

The Discussion is very interesting and points to useful future endeavours, with multiple output criteria to make the possible solution space smaller and get the right answer (outputs) for the right reason (see Kirchner 2006). As far as groundwater characterisation for rainfall-runoff modelling, the lag times may be very different for a "land" cell state and a "groundwater" cell state and potentially exceed one calendar year. Statistical correlation methods for groundwater variation with lag measured in months to years such as HARTT (e.g., Ferdowsian et al. 2001; Goodarzi 2020) are abstract methods that rely on variability of the assumed controlling mechanism, and have also been compared with neural network implementations. Whether these might provide some information or inspiration for additional work is unknown.

> **Response**: Splitting the regularization into short and long-term behavior is part of our inspiration for the next steps because it allows us to gain insight into the groundwater system using streamflow data, which has more abundant and consistent data.

**Reviewer references**

Ferdowsian R, Pannell DJ, McCarron C, Ryder A and Crossing L (2001) Explaining groundwater hydrographs: separating atypical rainfall events from time trends. Australian Journal of Soil Research, 39(4), 861–875, doi: 10.1071/SR00037

Goodarzi M (2020) Application and performance evaluation of time series, neural networks and HARTT models in predicting groundwater level changes, Najafabad Plain, Iran. Sustainable Water Resources Management, 6, 67, doi: 10.1007/s40899-020-00427-2

Kirchner, J. W. (2006) Getting the right answers for the right reasons: Linking measurements, analyses, and models to advance the science of hydrology. Water Resources Research, 42, W03S04, doi:10.1029/2005WR004362

Lienhard JH (1964) A statistical mechanical prediction of the dimensionless unit hydrograph. Journal of Geophysical Research, 69(24), 5231-5238, doi: 10.1029/JZ069i024p05231

Ocak A and Bayazit M (2003) Linear Reservoirs in Series Model for Unit Hydrograph of Finite Duration. Turkish Journal of Engineering and Environmental Science, 27(2), 107-113, https://search.trdizin.gov.tr/en/yayin/detay/31619/

Pilgrim DH, Chapman TG and Doran DG (1988) Problems of rainfall-runoff modelling in arid and semiarid regions. Hydrological Sciences Journal, 33(4), 379-400, doi: 10.1080/02626668809491261

Rodriguez-Iturbe I and Valdes JB (1979) The geomorphic structure of hydrologic response. Water Resources Research, 15(6), 1409-1420, doi: 10.1029/WR015i006p01409

Sherman LK (1932) Streamflow from rainfall by the unit-graph method. Engineering News Record, 108, 501–505.

**Response references**

De la Fuente, L. A., Gupta, H. V., and Condon, L. E.: Toward a Multi-Representational Approach to Prediction and Understanding, in Support of Discovery in Hydrology, Water Resources Research, 59, https://doi.org/10.1029/2021WR031548, 2023.

Jiang, S., Zheng, Y., Wang, C., and Babovic, V.: Uncovering Flooding Mechanisms Across the Contiguous United States Through Interpretive Deep Learning on Representative Catchments, Water Resources Research, 58, https://doi.org/10.1029/2021WR030185, 2022.